# The splicing factor kinase, SR protein kinase 1 (SRPK1) is essential for late events in the human papillomavirus life cycle

Arwa A. A. Faizo[¤a], Clare Bellward[¤b], Hegel R. Hernandez-Lopez[¤c], Andrew Stevenson, Quan Gu, Sheila V. Graham [iD]ᴼ*

MRC-University of Glasgow Centre for Virus Research, School of Infection and Immunity, College of Medical, Veterinary and Life Sciences, University of Glasgow, Glasgow, Scotland

ᴼThese authors contributed equally to the work.
¤a Current address: Special Infectious Agents Unit, King Fahd Medical Research Center, King Abdulaziz University, Jeddah, Saudi Arabia and Department of Medical Laboratory Technology, Faculty of Applied Medical Sciences, King Abdulaziz University, Jeddah, Saudi Arabia
¤b Current address: Fios Genomics, BioCube 1, Edinburgh BioQuarter, Edinburgh, Scotland
¤c Current address: Illumina Belgium, Mechelen, Belgium
* Sheila.Graham@glasgow.ac.uk

## Abstract

Human papillomaviruses (HPV) infect epithelia to cause benign lesions or warts. However, the so-called "high risk" HPVs infecting the anogenital region and the oropharynx can cause precancerous lesions that may progress to malignant tumours. Understanding the HPV life cycle is important to the discovery of novel antiviral therapies. HPV uses cellular splicing to produce the full suite of viral mRNAs. Members of the serine/arginine rich (SR) protein family can positively regulate splicing. SR protein activity and cellular location is regulated by phosphorylation of their serine-arginine domains. SR protein kinases (SRPK) can phosphorylate SR proteins. This licenses their nuclear entry and promotes nuclear splicing together with another SR protein kinase, Clk1. SRPIN340 is a specific inhibitor of SRPK1. It has been reported to inhibit replication of HCV, Sindbis virus and HIV. We show here that SRPIN340 inhibits the expression of the viral replication and transcription factor, E2. Loss of HPV E4 and L1 late proteins was also observed. RNA sequencing showed SRPIN340 treatment resulted in gene expression changes opposite to those induced by HPV16 infection. In particular, the loss of the epithelial barrier was restored. SRPIN340 treatment led to changes in alternative splicing of 935 RNAs and pathway analysis showed a predominance of changes to RNAs encoding proteins involved in chromatin conformation, DNA repair and RNA processing. Short term SRPIN340 treatment (two to three days) was not associated with changes in proliferation or differentiation of keratinocytes. Since SRPK1 controls the E2 viral replication and transcription factor, targeting this kinase, or the phosphorylation events it mediates, could be considered as a therapeutic strategy for HPV16 infection.

**Data availability statement:** The raw data for the results described here is available at https://doi.org/10.5525/gla.researchdata.1876. RNA-Seq data sets are freely available from the European Nucleotide Archive accession number PRJEB80489.

**Funding:** Studentship # K1513 from King Abdulaziz University, Jeddah, to author AAA. Faizo Medical Research Council grant # MC_UU_00034/5 supporting author Q.Gu https://www.ukri.org/councils/mrc/ The funders played no role in the study design, data cllection and analysis. decision to publish or preparation of the manuscript.

**Competing interests:** The authors have declared that no competing interests exist.

## Author summary

Human papillomavirus (HPV) can cause precancerous and cancerous tumours of the anogenital region, for example, the cervix. HPV infects epithelial cells and utilizes the host cell gene expression machinery to produce its own proteins. Splicing is a key mechanism used by HPV to synthesise messenger RNAs encoding the viral proteins. Splicing is positively controlled by SR proteins. These proteins enter the nucleus and regulate splicing upon phosphorylation by SR protein kinase (SRPK). SRPIN340 is a specific inhibitor of SRPK1. SRPIN340 treatment of HPV16-infected epithelial cells resulted in loss of expression of the viral replication and transcription factor, E2 and abrogation of late events in the viral life cycle. SRPIN340 treatment counteracted some gene expression changes to the epithelium known to be cause by HPV infection. These data suggest that SRPK1 is essential for the HPV life cycle. It may have potential as an anti-viral target.

## Introduction

Human papillomaviruses (HPVs) are sexually transmitted. They infect cutaneous and mucosal epithelia to cause benign lesions or warts [1]. Most HPV infections are asymptomatic and are eventually cleared by the host immune system [2]. However, persistent infection with one of the so-called high-risk HPVs (HR-HPVs) can cause cancers in the oropharyngeal and anogenital mucosa. Globally, infections with HR-HPVs 16 and 18 account for up to 70% of cervical cancers and up to 40% of head and neck cancers [3]. Vaccination against HR-HPVs is now delivered to young people prior to sexual debut [4]. However, the vaccines are prophylactic. Thus there are several decades-worth of infected individuals who have not been vaccinated or have declined vaccination. Development of antivirals against HR-HPV infection is essential. Understanding viral gene regulation is important to this aim.

The HPV life cycle is intimately linked to differentiation of the epithelium it infects. This linkage is tightly regulated by virus-host interactions [5]. HPV early proteins E1 and E2 control viral replication. They are first expressed in basal epithelial cells to establish infection. They are expressed later in the upper epithelial layers to enable vegetative viral genome replication prior to virion formation [6]. E2 is also the viral transcription factor. Viral E6 and E7 proteins are expressed throughout the life cycle. They activate the cell cycle and stop the apoptotic response by degrading p53 [7,8]. E4, the most abundant late protein, can alter keratinocyte filaments to facilitate virus replication and egress [9]. The final event in the HPV life cycle is expression of the capsid proteins L1 and L2. Capsid protein expression is restricted to the outermost epithelial layers, an immune-privileged site, to avoid activation of the host immune response [5].

Constitutive and alternative splicing are essential host cell mechanisms for production of multiple mature mRNAs from a single gene. Splicing is regulated and enhanced by serine-arginine-rich splicing factors (SRSFs or SR proteins). They control splice site selection, exon definition, spliceosome recruitment and stabilisation [10,11]. They can bind to exonic and intronic sequence enhancers (ESEs/ISEs) to promote splicing at specific sites. The classical SR proteins include SRSFs1-9. All SRSFs contain an N-terminal RNA recognition motif (RRM) and a C-terminal serine-arginine-rich (RS) domain [11]. Some SR proteins (e.g. SRSF2, SRSF3 and SRSF9) have one RRM but for others, (e.g. SRSF1) there is an additional pseudo-RRM [12]. SR proteins are present at steady state in the nucleus. However, some SR proteins (e.g. SRSF1, SRSF3, and SRSF7) can shuttle between the nucleus and cytoplasm [13]. Therefore, there are different SR protein subclasses.

The RS domain of SR proteins is subject to phosphorylation by kinases including serine/arginine protein kinases (SRPK) and CDK-like kinase 1 (Clk1) [14,15]. This post-translational modification regulates both the function and subcellular localisation of SRSFs [16,17]. SRPK is normally present in the cytoplasm of cells where it phosphorylates newly synthesised SRSFs, to licence their entry into the nucleus [18]. There are three SRPKs in human cells (SRPK1, 2 and 3). Only SRPK1 and SRPK2 are expressed in epithelial cells. In the nucleus, SRPK1 interacts with Clk1 to promote splicing [15]. Data have shown that the SRSF1 pseudo-RRM can regulate the directionality of phosphorylation [19]. This suggests that SR proteins containing two RRMs may be phosphorylated and control splicing in a different manner from their single RRM–containing counterparts.

HPV gene expression is regulated at the transcriptional and post-transcriptional levels [20]. Production of viral proteins occurs from multiple alternatively spliced mRNAs [21]. HPV utilizes the host cell splicing machinery, including SR proteins, for viral protein expression. There is evidence that HPV controls host cell splicing. For example, HPV16 and HPV31 E2 protein can transcriptionally activate the promoters of the genes encoding SRSFs 1, 2 and 3, up-regulates SRPK1. It can also can bind SRSFs [22–24].

SRPIN340 is a specific inhibitor of SRPK1 through binding to the enzyme's active site [25,26]. SRPIN340 possesses anti-angiogenesis properties by inhibiting VEGF splicing. It has been suggested as a potential therapy against exudative age-related macular degeneration and diabetic nephropathy, among other diseases [25,26]. SRPIN340 can inhibit replication of HIV, Sindbis virus and hepatitis C virus [27,28]. SRPK1 and SR proteins are essential factors for the life cycle of viruses such as herpes simplex virus type 1 (HSV1) [29] and hepatitis B virus (HBV) [30]. This suggests that drugs to inhibit SRPK1 could have a general activity against a range of viruses.

We tested the hypothesis that SRPIN340 could inhibit the HPV life cycle through regulating the SR proteins required for HPV mRNA expression [31, 32]. We focused on HPV16 as the most prevalent HPV genotype. We assessed the effect of SRPIN340 on viral protein expression (E2, E4, E6, E7 and L1). We also investigated whether SRPIN340 is associated with toxicity to the host cells. Our results demonstrate loss of the HPV E2 replication/transcription factor expression upon SRPIN340 inhibition of SRPK1. A reduction in HPV E4 and L1 late protein levels was also observed. Early gene expression was unaffected. No significant change to epithelial growth or differentiation or to cellular proliferation or viability was detected upon short term drug treatment (two to three days). RNA sequencing revealed SRPIN340-induced changes to keratinocyte differentiation, and to the epithelial barrier, opposite to those caused by HPV infection.

## Results

Work by others has shown inhibition of replication of HIV, Sindbis virus and hepatitis C virus by the SRPK1 inhibitor SRPIN340 [27,28]. Previously we showed HPV16 E2-mediated up-regulation of SRPK1 during the HPV16 life cycle [23]. Now we wished to determine if SRPIN340 could act as an antiviral against HPV. First, we wanted to check if there was any effect of the drug on the growth or morphology of 3D raft tissues supporting HPV infection. HPV16 was chosen as the primary focus due to its high prevalence among high-risk HPV types associated with anogenital and oropharyngeal cancers. We grew 3D raft tissues using HPV16-infected keratinocytes (NIKS16 cells: normal immortalised keratinocytes stably transfected with HPV16 genomes [33,34]) for 14 days. At this time the tissues were differentiated. We chose to test concentrations of the drug at 10 μM or 50 μM since these concentrations had been optimised in previous virus replication studies [27,28]. Preliminary titration assays showed these concentrations were optimal for effective inhibition of SRPK1. Compared to the drug vehicle DMSO, the addition of SRPIN340 in the culture medium after 12 days of tissue

growth at either 10 μM or 50 μM, and applied for 48 hours, caused disordered growth of the raft tissues (Fig 1A). Next, we developed a method of applying the drug to the upper surface of the tissues in a 50 μl droplet also for the last 48 hours of a 14-day growth period. This volume was chosen because it covered the surface of the tissue without reaching the edge. In contrast to the significant disruption to tissue morphology when SRPIN340 was added to the culture media, there was no observable difference in morphology between DMSO or 10 μM SRPIN340-treated tissues when the drug was applied to the top of the tissues (Fig 1B). Some disruption of the upper epithelial layers was observed when the drug was applied at 50 μM (Fig 1B).

To test whether SRPIN340 could enter the epithelium upon topical application we applied the fluorophore Alexa 488 in a 50 μl droplet to the top of the raft tissues. This molecule has a similar chemical structure to SRPIN340, whose molecular weight is 349 Daltons. Alexa Fluor 488 with a molecular weight of 570 Daltons is known to be transmitted between cells by gap junctions. Gap junctions permit intercellular spread of molecules >1 kDa in size [35]. As a control, topical application of DMSO produced no autofluorescence (Fig 1C). Topical application of Alexa 488 revealed penetration down to the basal layers of the tissues. Fluorescence was distributed into all the cell layers of the epithelium (Fig 1D). This suggests that SRPIN340, a smaller molecule than Alexa Fluor 488, could travel through gap junctions throughout the epithelium.

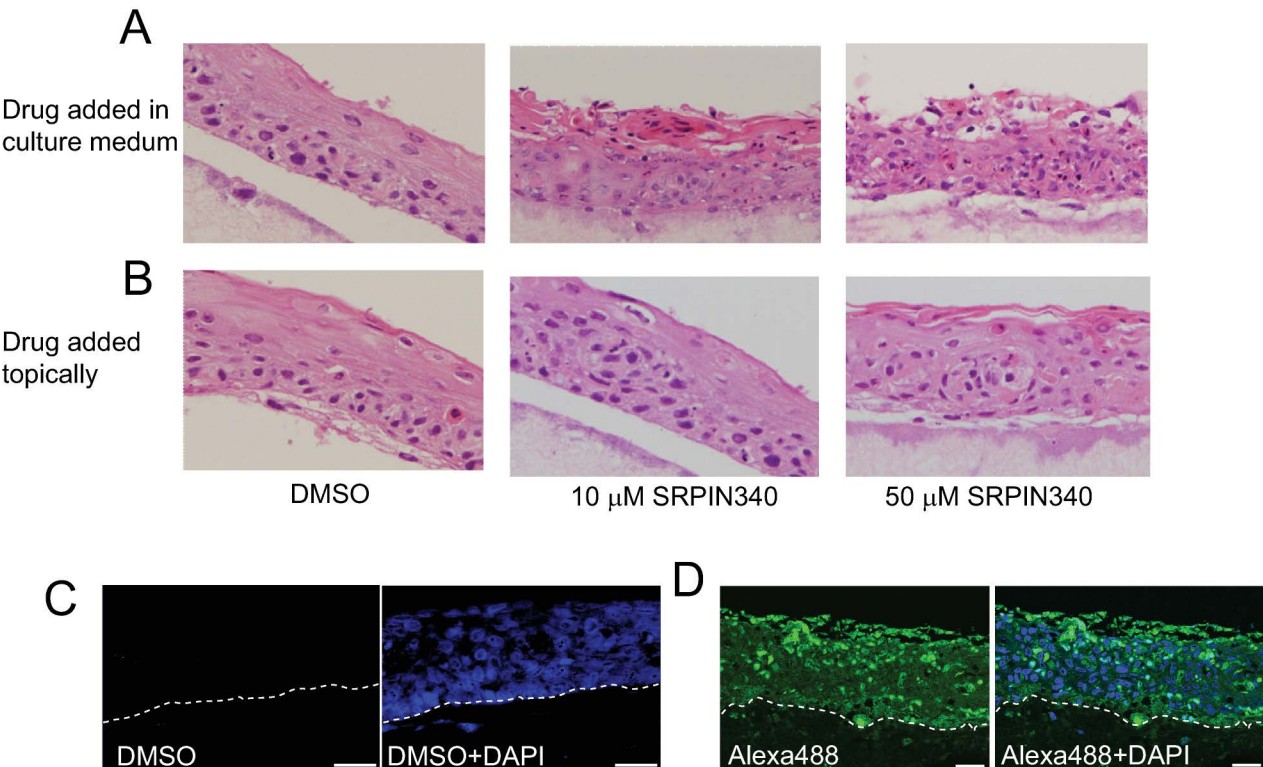

**Fig 1. Topical SRPIN340 treatment of NIKS16 3D raft tissues does not disrupt tissue morphology and may be transmitted throughout the tissues.** (A) H&E staining of NIKS16 3D raft tissue sections cultured for 12 days then grown in the presence of 10 or 50 μM SRPIN340 in the culture medium for 48 hours. (B) H&E staining of NIKS16 3D raft tissue sections cultured for 12 days followed by topical application of 10 or 50 μM SRPIN340 in a 50 μl droplet and incubation for 48 hours. (C) Autofluorescence control of NIKS16 3D cultured raft tissues stained with the same secondary antibody used in (D). (D) Immunofluorescence microscopy of NIKS16 3D cultured raft tissues topically treated with Alexa-fluor 488 (Alexa488: green staining). Tissues in the right-hand panels of (C) and (D) are counterstained with DAPI. The junction of the basal layer of cells with the dermal equivalent is indicated by a dotted line. Size bars = 50 μM.

## SRPIN340 treatment leads to reduced phosphorylation and relocation to the cytoplasm of shuttling SR proteins in keratinocytes

Next, we used NIKS16 cells differentiated in monolayer culture to determine if topically applied SRPIN340 had biochemical effects on SR proteins. These cells contained episomal HPV16 genomes (S1A Fig). They expressed differentiation markers keratin 10 and involucrin (S1B Fig). The cells expressed HPV16 E2 replication/transcription factor and viral capsid protein L1 (S1B Fig). We quantified the levels of phosphorylated SRSFs1, 2 and 3 in NIKS16 cells differentiated in monolayer culture for eight days then topically treated for 48 hours with either DMSO or SRPIN340 at 10 μM. SRSF1 and SRSF3 are substrates of SRPK1. SRSF2 is a poor substrate of SRPK1 but can be phosphorylated by SRPK2. Using a phospho-specific antibody (Mab104) we found reduced levels of phosphorylated SRSF1 (Fig 2A) and SRSF3 (Fig 2C) upon drug treatment. Levels of phosphorylated SRSF2 did not change significantly, as expected (Fig 2B). Next, we analysed the effect of SRPIN340 on the subcellular location of

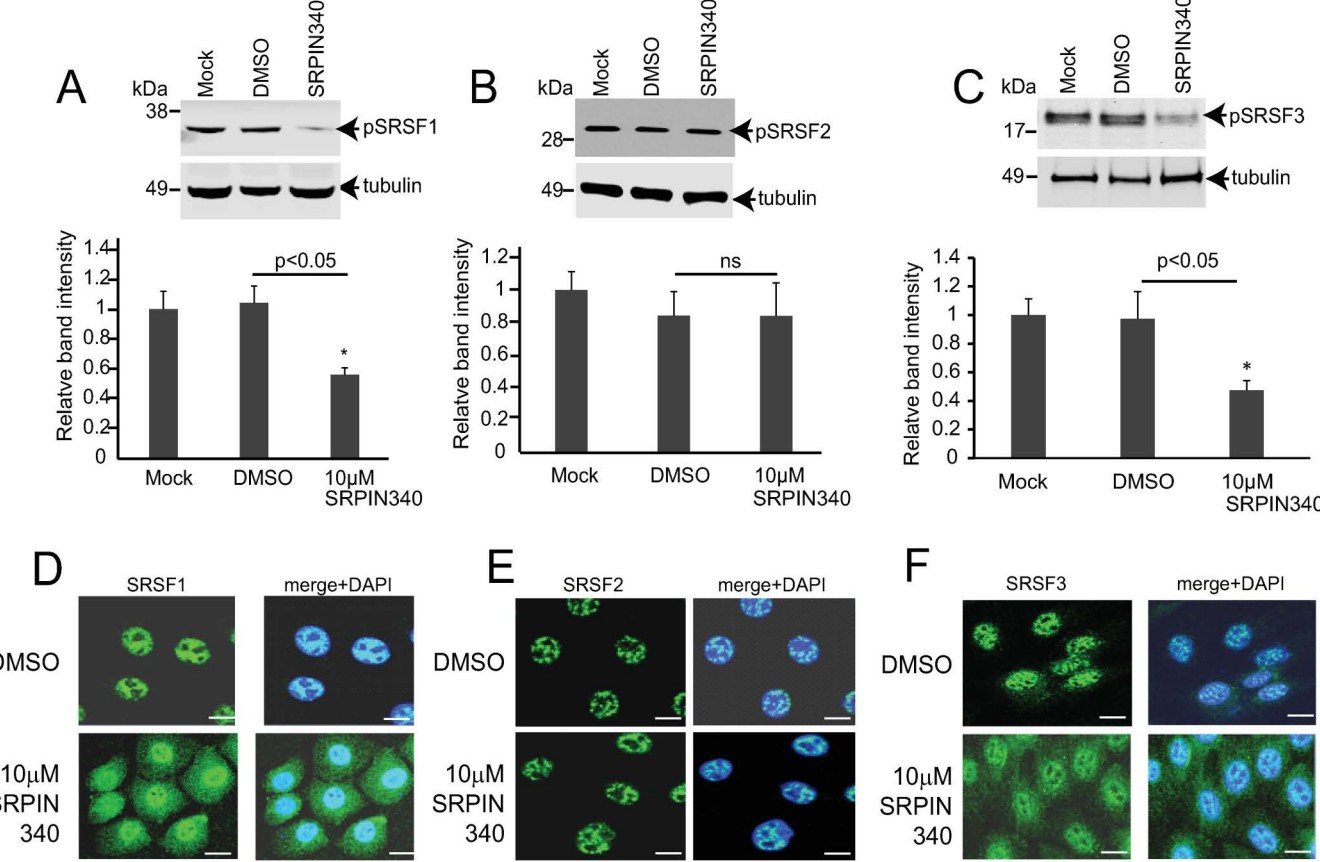

**Fig 2. SRPIN340 inhibits phosphorylation of SRSF1 and SRSF3.** Differentiated NIKS16 cells in monolayer culture were mock-treated or treated with 10 μM SRPIN340 or with the drug vehicle, DMSO. (A) Representative western blot and graph of the quantification of levels of phosphorylated SRSF1 relative to levels of β-tubulin. (B) Representative western blot and graph of the quantification of levels of phosphorylated SRSF2 relative to levels of β-tubulin. (C) Representative western blot and graph of the quantification of levels of phosphorylated SRSF3 relative to levels of β-tubulin. The data shown are the mean and standard deviation from the mean from three separate experiments. p<0.05, p-value less than 0.05. *, statistically significant difference in relative protein levels. ns, not statistically significant. (D-F) NIKS16 cells were differentiated in monolayer culture and treated with DMSO or with 10 μM SRPIN340. (D) Immunofluorescence microscopy of cells stained with an antibody against SRSF1. (E) Immunofluorescence microscopy of cells stained with an antibody against SRSF2. (F) Immunofluorescence microscopy of cells stained with an antibody against SRSF3. The left-hand panels show SRSF staining in green. The right-hand panels show SRSF staining merged with DAPI (blue staining). Representative images from three separate experiments are shown. Size bars = 10μm.

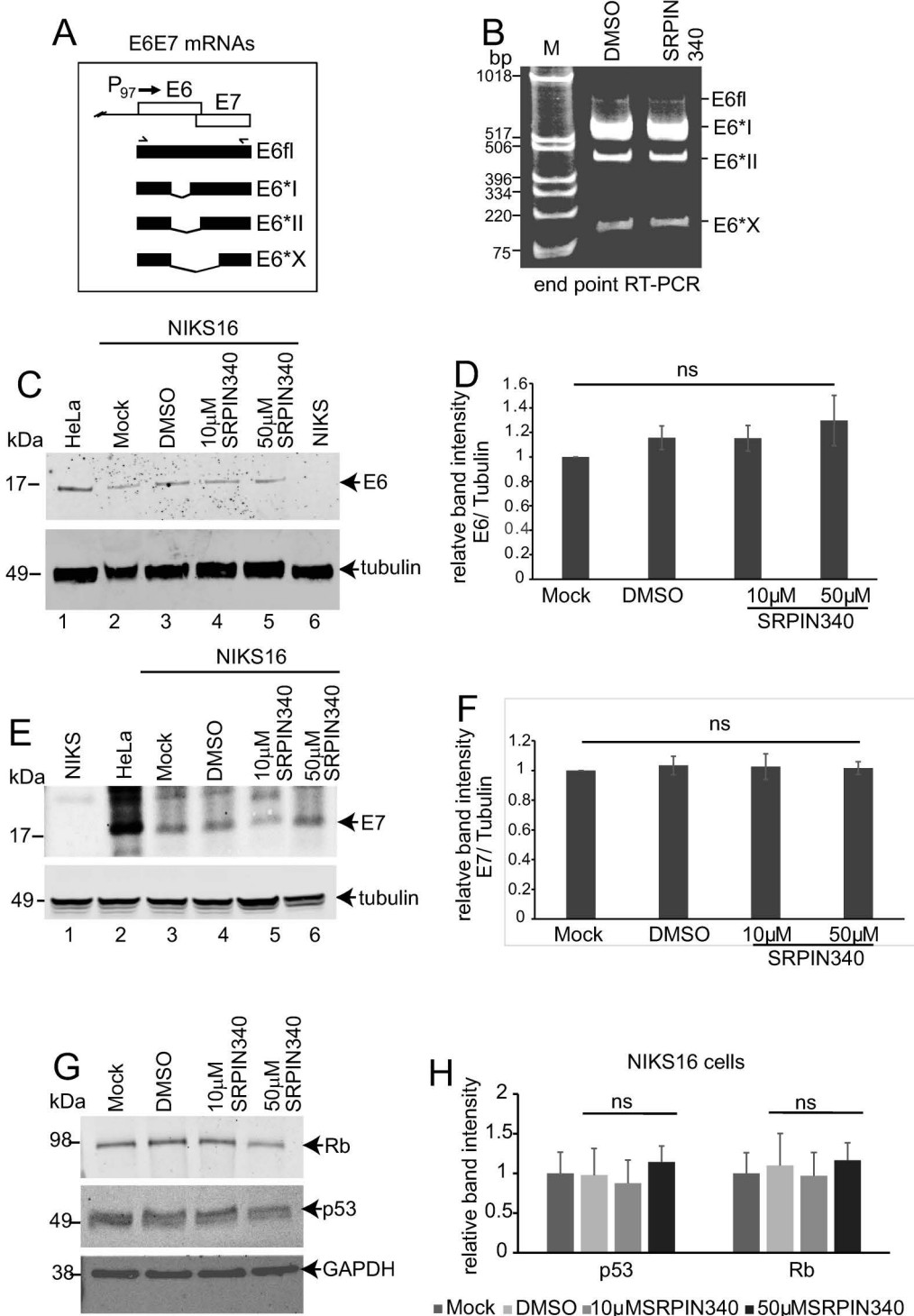

**Fig 3. SRPK1 inhibition does not alter HPV16 oncoprotein expression or activity.** (A) Diagram of the E6E7 coding region of the HPV genome showing the unspliced (E6fl) and three spliced RNAs transcribed from the region (E6*I, E6*II, E6*X). $P_{97}$, HPV16 early promoter located at nucleotide 97 on the genome. Forward and reverse facing chevrons indicate the approximate position of the primers used in the PCR reaction in (B). (B) Gel image showing end point RT-PCR products using the primers indicated in (A) to amplify all four E6E7 mRNAs from NIKS16 cells differentiated in monolayer culture and treated with DMSO or 10 μM SRPIN340. M = size marker. Western blot of levels of (C) E6 and (E) E7 in monolayer cultured and differentiated NIKS16 cells either mock-treated or treated with

DMSO, or SRPIN340 at 10 μM or 50 μM. Differentiated HPV-negative NIKS cells extracts were included in each blot to show specificity of the antibodies. Cell extracts from HeLa cells grown in monolayer culture were included on the blot as a positive control for detection of E6 and E7. The antibodies against E6 and E7 detect both HPV16 and HPV18 oncoproteins. The upper portion of each blot was reacted with an anti β-tubulin antibody as a loading control. Graphs showing quantification of three separate western blot experiments of the levels of (D) E6 and (F) E7 relative to β-tubulin, in NIKS16 cells differentiated in monolayer culture using the various experimental conditions in the western blots. (G) Western blots of levels of Rb, p53 and GAPDH as a loading control, in differentiated monolayer-cultured NIKS16 cells mock-treated or treated with DMSO, or SRPIN340 at 10 μM or 50 μM. H. Graph showing quantification of p53 and Rb levels in the western blot in (G). Expression levels were calculated relative to the GAPDH loading control. The data in all the graphs are the mean and standard deviation from the mean from three separate experiments. ns, not statistically significant.

SRSF1 and SRSF3. The import of these SR proteins into the nucleus is controlled by SRPK1 phosphorylation. SRSF2 is mostly confined to the nucleus [10,11]. Compared to a mainly nuclear location in DMSO-treated NIKS16 cells, SRSF1 and SRSF3 were found in both the nucleus and the cytoplasm in SRPIN340-treated NIKS16 cells (Fig 2D,F). In contrast, SRSF2 maintained a nuclear location in both DMSO and SRPIN340-treated cells (Fig 2E). These data confirm efficacy of SRPIN340 in inhibiting the activity of SRPK1 in in HPV16-positive keratinocytes.

## SRPIN340 treatment does not change E6 or E7 oncoprotein expression or activity

First, as a measure of HPV early gene expression, we investigated if SRPIN340 affected the expression of the E6 and E7 alternatively spliced mRNAs (Fig 3A). For these experiments we used NIKS16 cells grown in monolayer culture. End-point PCR analysis revealed the four main splice isoforms of E6E7 mRNAs. Their levels were unchanged in the presence of the drug (Fig 3B). Next, we investigated E6 and E7 protein levels in drug-treated NIKS16 cells and in control, HPV-negative NIKS cells. HeLa cells were used as a positive control since they express high levels of E6 and E7 allowing unambiguous identification of E6 and E7 protein bands. The antibodies used detect both HPV16 and HPV18 oncoproteins. E6 was detected in HeLa cells (Fig 3C lane 1) but was not detected in NIKS cells (Fig 3C, lane 6). E6 was present at similar levels in all NIKS16 lanes (Fig 3C, lanes 2-5). E7 was detected in HeLa cells (Fig 3E, lane 2) but was not detected in NIKS cells (Fig 3E, lane 1). E7 was present in all NIKS16 lanes (Fig 3E, lanes 3-6). There was no significant difference in levels of E6 (Fig 3D) or E7 proteins (Fig 3F) in NIKS16 cells upon drug treatment. Expression levels of the E6 degradation target protein, p53 and the E7 target, Rb were also unaffected comparing mock-treated to drug treated NIKS16 cells (Fig 3G, H). The data suggest short term treatment with SRPIN340 has no oncogenic effect.

## SRPIN340 inhibits the expression of the viral E2 replication/transcription factor

Next, we investigated the expression of viral late mRNAs. We used NIKS16 cells grown and differentiated in monolayer culture to express viral late proteins (S1B Fig). Expression of the major late HPV16 spliced transcript, E1^E4^L1 and the L2L1 portion of the readthrough transcript E1^E4, E5, L2, L1 (Fig 4A) remained present following SRPIN340 treatment in differentiated monolayer cultured NIKS16 cells (Fig 4B,C).

For the following experiments, cells were grown in monolayer for six days before drug treatment for 48 hours since E2 is expressed in keratinocytes that are not fully differentiated. Analysis of the major mRNA encoding the viral E2 replication/transcription factor [36] (Fig

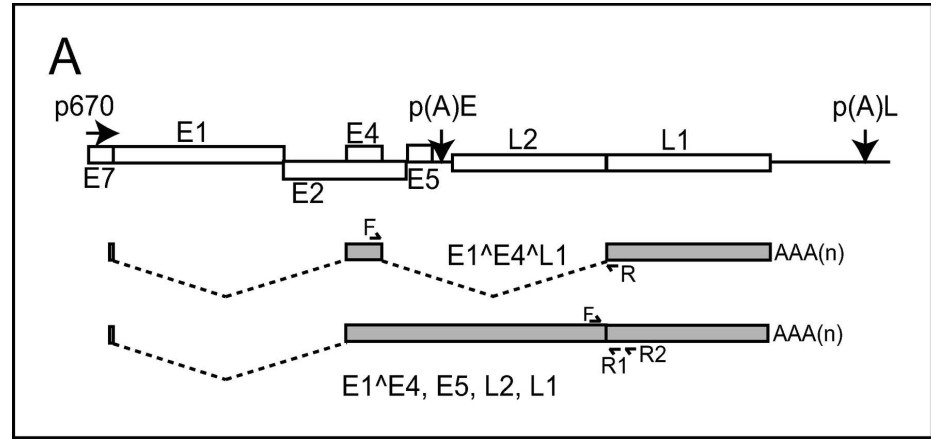

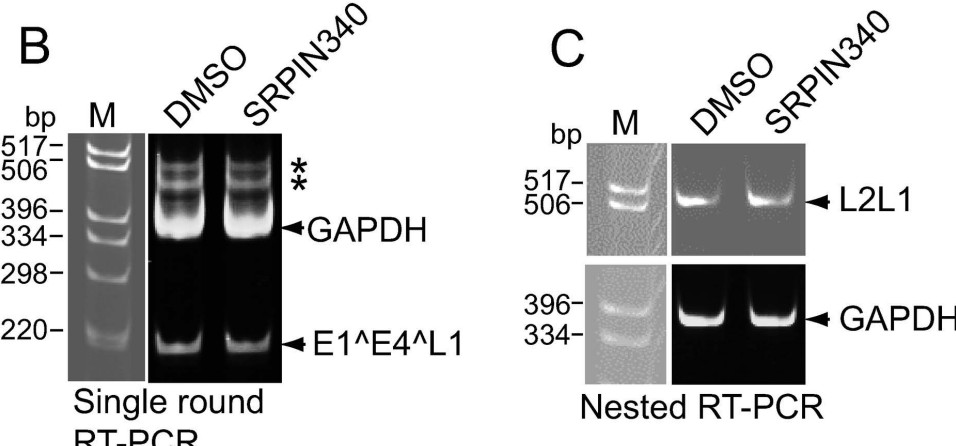

**Fig 4. SRPIN340 treatment does not alter expression of the HPV16 major late mRNAs.** (A) Diagram of the portion of the HPV16 genome downstream of the $P_{670}$ late promoter located in the E7 gene region. The E1^E4^L1 alternatively spliced mRNA encodes E4 and L1 and the readthrough mRNA E1^E4, E5, L2 and L1 is thought to encode L2. These RNAs are diagrammed below the genome map. $P_{670}$, HPV16 late promoter located at nucleotide 670. p(A) E, early polyadenylation site, p(A)L, late polyadenylation site. Open rectangles, HPV16 genes E1, E2, E4, E5, L2 and L1. Shaded rectangles, open reading frames in the RNAs. Dotted lines represent RNA sequences spliced out to form the mRNAs. AAAn, poly(A) tail. Chevrons indicate the approximate positions of forward and reverse primers used in the PCR reactions shown in (B) and (C). (B) Ethidium bromide-stained gel showing the PCR products obtained from amplification of cDNA prepared from monolayer cultured and differentiated NIKS16 cells using the primer pair indicated in (A) for the E1^E4^L1 mRNA. GAPDH primer pairs were added to the same reaction for a loading control. * non-specific bands. (C) Upper panel: ethidium bromide-stained gel showing the PCR products obtained from amplification of cDNA prepared from monolayer cultured and differentiated NIKS16 cells using the primer pairs indicated in (A) in a nested PCR reaction to amplify across the L2-L1 junction from the E1^E4, E5, L2, L1 readthrough mRNA. Lower panel: amplification of GAPDH as a loading control. In (B) and (C) the gel pictures are split because additional reactions were electrophoresed between the marker lane and the DMSO and SRPIN340 lanes.

) was inhibited when differentiated NIKS16 cells were treated with the drug (Fig 5B, lane 3). Treatment with DMSO (Fig 5B, lane 1) or with SRPIN349, which does not inhibit the kinase activity of SRPK1 [28] (Fig 5B, lane 2), did not affect the expression of the E2 mRNA. We carried out western blot analysis of SRPIN340-treated NIKS16 cells differentiated for eight days in monolayer culture. This revealed reduced expression of the E2 protein in the presence of the drug (Fig 5C). Protein extract from HPV-negative NIKS cells, grown and differentiated in the same manner, was included in the western blot as a control for the specificity of the

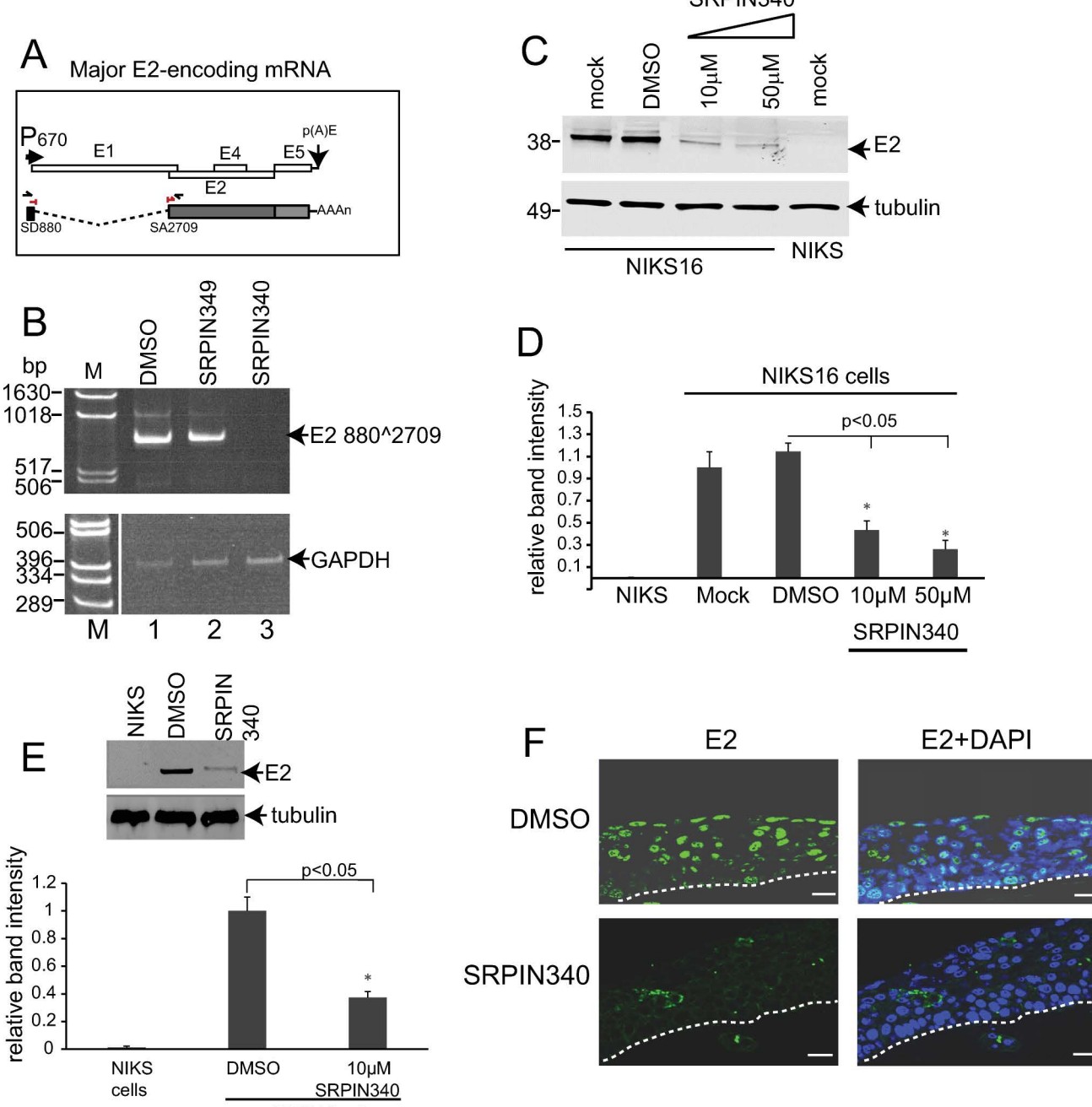

**Fig 5. SRPK1 inhibition leads to loss of the spliced mRNA encoding E2.** (A) Diagram of the early region of the HPV16 genome downstream of the E6E7 gene region. $P_{670}$, HPV16 late promoter located at nucleotide 670 in the E7 coding region. p(A)E, the HPV16 early polyadenylation site. Open rectangles, HPV16 early genes E1, E2, E4, E5. Below is the genome map is shown a diagram of the major spliced mRNA encoding E2. Shaded rectangles, open reading frames. Dotted lines represent RNA sequences spliced out to form the E2 mRNA. AAAn, poly(A) tail. Black chevrons indicate forward and reverse primers used in the first round of the nested RT-PCR reaction shown in (B). A red split chevron indicates the forward primer used in the second round of the nested PCR amplification shown in (B). This cross-splice epitope primer was used with the reverse primer located inside the E2 open reading frame. SD880, splice donor site at nucleotide 880. SA2709, splice acceptor site at nucleotide 2709. (B) Upper panel: ethidium bromide-stained gel showing the products of an RT-PCR reaction to amplify the E2 mRNA from NIKS16 cells differentiated in monolayer and treated with DMSO (lane 1) or with SRPIN349 (lane 2), not inhibitory to SRPK1, or with SRPIN340 (lane 3). Lower panel: GAPDH cDNA was amplified as a loading control. The gel picture is split because the GAPDH reactions were electrophoresed on the right-hand side of the same gel used to visualise the E2 amplification products. These were electrophoresed on the left-hand side adjacent to the size markers. (C) Western blot showing levels of E2 in monolayer-cultured and partially differentiated NIKS16 cells (differentiated for six days) mock-treated, DMSO-treated or treated with either 10 μM

or 50 μM SRPIN340 for 48 hours. Protein extract from HPV-negative differentiated monolayer-cultured NIKS cells was included in the experiment to demonstrate specificity of the E2 antibody. β-tubulin was used as a loading control. (D) Graph showing quantification of E2 levels relative to the β-tubulin loading controls in the various experimental conditions used in the western blot in (C). The data shown are the mean and standard deviation from the mean from quantification of three separate western blot experiments. $p < 0.05$*, p-value indicating a significant difference in E2 expression levels. (E) Graph showing quantification of E2 levels relative to the β-tubulin loading controls using protein extracts from NIKS16 cells grown in differentiated 3D raft tissue cultures. The data shown are the mean and standard deviation from the mean from three separate western blot experiments Representative western blots showing E2 and β-tubulin levels are shown above the graph. $p < 0.05$*, p-value indicating a significant difference in E2 expression levels. (F) Immunofluorescence staining with an E2 antibody (green staining, left hand panels) and counterstained with DAPI (blue staining, right hand panels) of sections of differentiated 3D cultured NIKS16 raft tissues topically treated with DMSO or with 10 μM SPRIN340. Dotted lines indicate the basal membrane of the tissues. Size bars=50 μm.

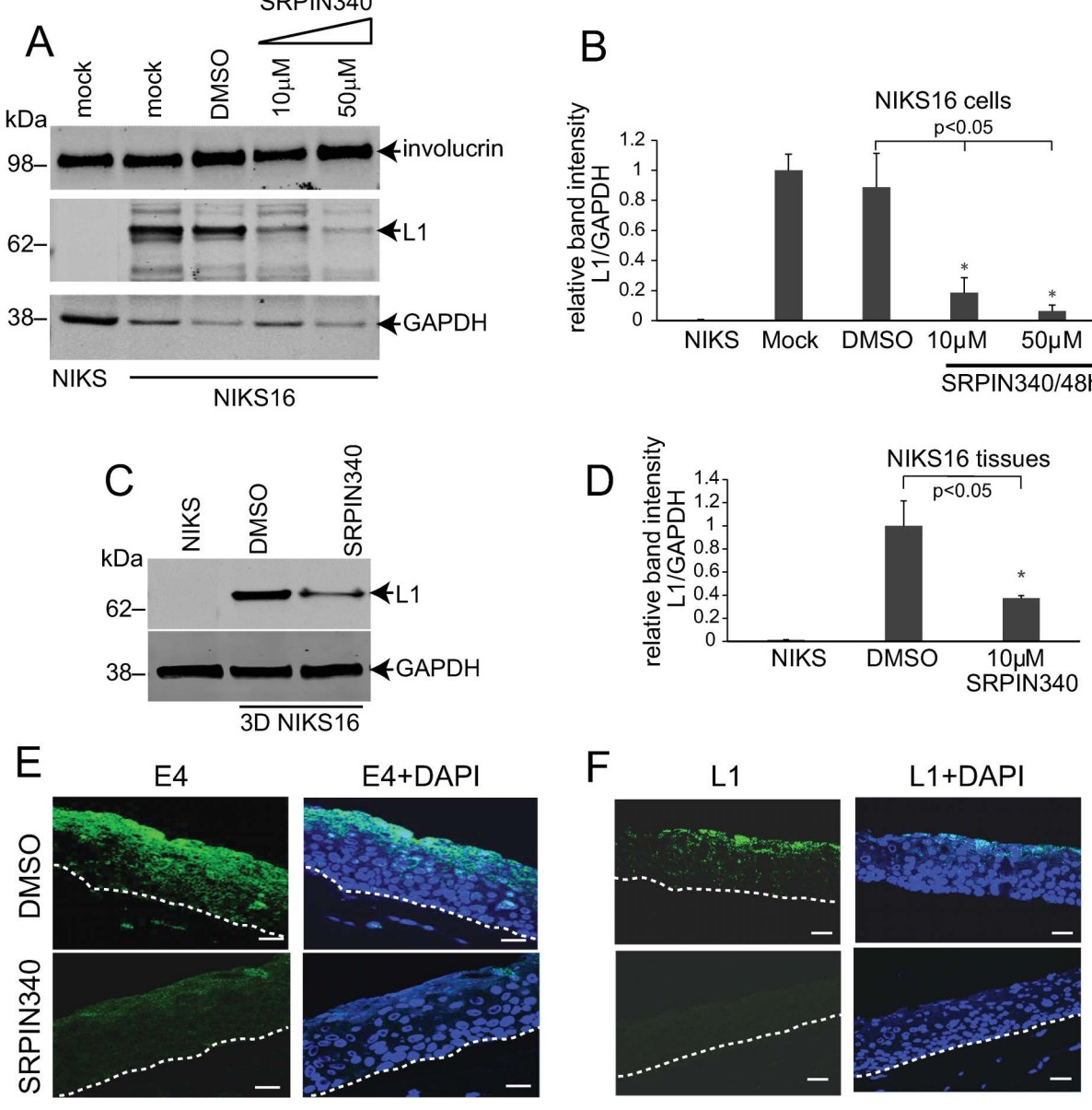

**Fig 6. SRPK1 activity is required for HPV16 late gene expression.** (A) Western blot analysis of levels of L1 protein in monolayer cultured and differentiated NIKS16 cells mock-treated or DMSO-treated or treated with 10 μM or 50 μM SRPIN340 for 48 hours. Protein extract from monolayer cultured and differentiated HPV-negative, NIKS cells was included on the blot as a control for the specificity of the L1 antibody. The upper portion of the blot was reacted with an antibody against involucrin to demonstrate cell differentiation. GAPDH

was used as a loading control. (B) Graph showing quantification of L1 levels relative to the GAPDH loading controls in the various experimental conditions used in the western blot in (A). The data shown are the mean and standard deviation from the mean from three separate western blot experiments. p<0.05*, p-value indicating a significant difference in L1 expression levels. (C) Western blot analysis of levels of L1 protein in differentiated 3D raft cultured NIKS16 tissues topically-treated with either DMSO or 10 μM SRPIN340. HPV-negative NIKS differentiated 3D raft cultured tissue extract was included on the blot as a control for the specificity of the L1 antibody. GAPDH was used as a loading control. (D) Graph showing quantification of L1 levels relative to the GAPDH loading control in the experimental conditions used in the western blot in (C). The data shown are the mean and standard deviation from the mean from three separate western blot experiments. p<0.05*, p-value indicating a significant difference in L1 expression levels. (E) Immunofluorescence microscopy of sections of NIKS16 differentiated 3D cultured raft tissues topically-treated with DMSO or with 10 μM SRPIN340 and stained with an antibody against E4 (green staining, left-hand panels) or counterstained with DAPI (blue staining, right-hand panels). (F) Immunofluorescence microscopy of sections of differentiated NIKS16 3D cultured raft tissues treated with DMSO or treated with 10 μM SRPIN340 and stained with an antibody against L1 (green staining, left-hand panels) or counterstained with DAPI (blue staining, right-hand panels). Dotted lines indicate the basal membrane of the tissues. Size bars=50 μm.

E2 antibody. Quantification of three separate experiments showed a statistically significant decrease in E2 levels upon drug treatment (Fig 5D). E2 levels were also significantly reduced in drug-treated NIKS16 3D differentiated raft tissues compared to DMSO-treated controls (Fig 5E). Confocal immunofluorescence microscopy confirmed E2 expression in the upper layers of NIKS16 3D differentiated raft cultures treated with the drug vehicle DMSO. However, very little E2 was present in SRPIN340-treated tissues (Fig 5F). Taken together, these data indicate that SRPIN340 treatment is associated with reduced expression of the spliced HPV16 E2 mRNA. This suggests that SRPK1 kinase activity is important for E2 expression.

## SRPIN340 inhibits HPV16 late protein expression

Loss of the viral replication/transcription factor E2 could result in abrogation of late events in the viral life cycle [6]. We found no change in mRNAs encoding viral late proteins (Fig 4). Now we examined late protein expression. Expression of L1, the major capsid protein, was significantly reduced by SRPIN340 treatment in differentiated NIKS16 cells grown in monolayer culture (Fig 6A,B). Differentiated HPV-negative NIKS cell extract was included in the western blot as a control for the specificity of the L1 antibody. Detection of involucrin revealed the cells to be differentiated. A reduction in L1 protein expression was also found in NIKS16 raft tissues grown for 12 days then topically treated with DMSO or SRPIN340 for 48 hours (Fig 6C,D). E4 is the most abundantly expressed late viral protein. Although antibodies against E4 work poorly in western blot, the E4 B11 antibody can be used successfully in confocal immunofluorescence microscopy. We performed immunofluorescence staining with E4 (Fig 6E) and L1 (Fig 6F) antibodies on sections of DMSO-treated or SRPIN340-treated NIKS16 differentiated 3D raft tissues. Data showed a substantial loss of both these late proteins in 3D differentiated raft tissues upon drug treatment. This suggests that SRPIN340 treatment is associated with inhibition of late protein expression.

## SRPIN340 results in significant changes in gene expression in NIKS16 cells opposite to those induced by HPV16 infection

SRPK1 inhibition can change how SR proteins regulate splicing, leading to changes in the cellular transcriptome. To our knowledge, SRPIN340-induced changes to cellular transcription in keratinocytes have never been reported. In addition, HPV E2 is known to regulate the expression of many cellular genes [37], including SR proteins [22]. Therefore, loss of E2 expression could contribute to SRPIN340-induced gene expression changes in HPV-infected cells. We carried out RNA sequencing (RNA-Seq) to compare gene expression profiles of SRPIN340-treated to DMSO-treated NIKS16 cells grown in monolayer culture in three

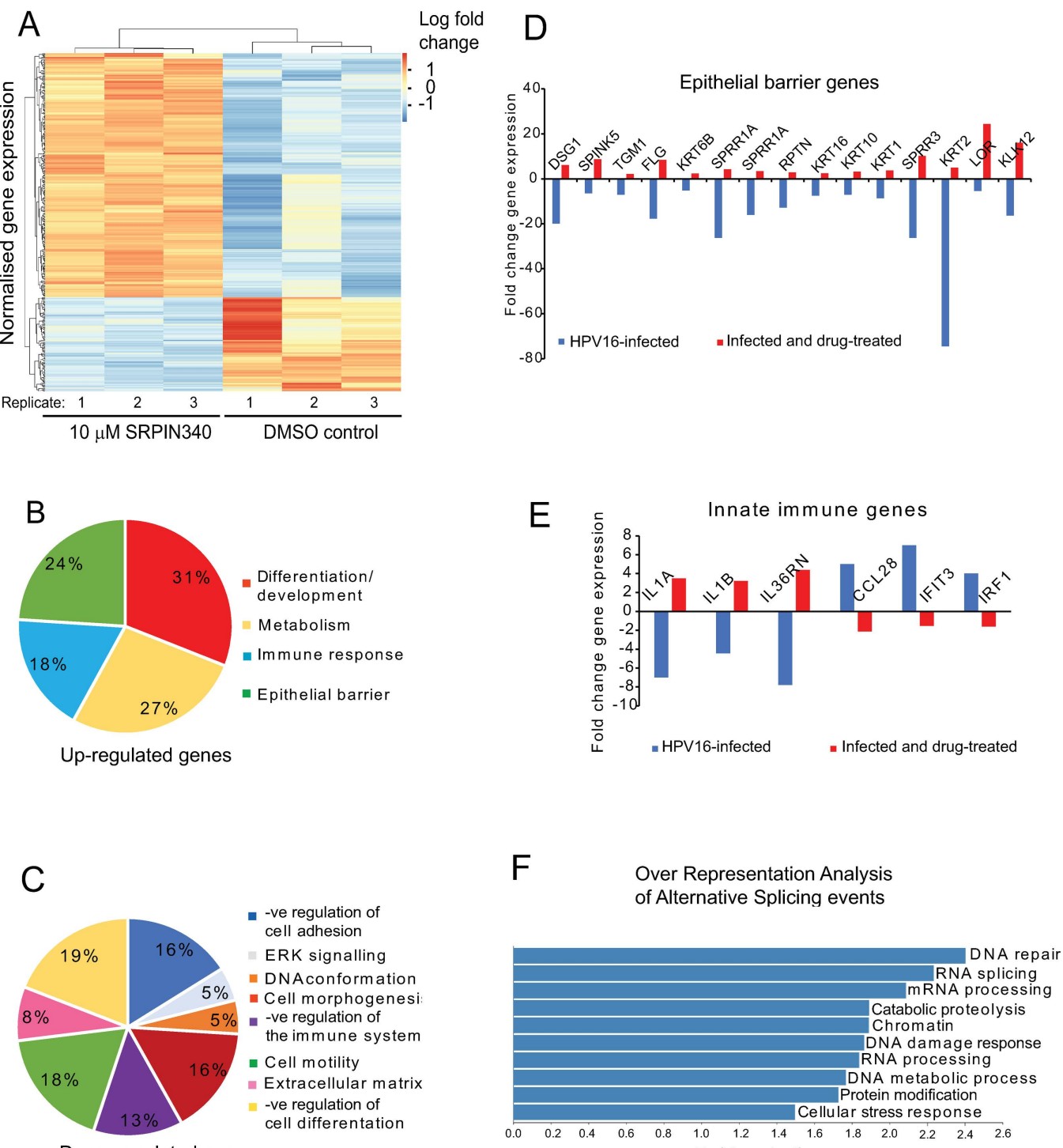

**Fig 7. RNA sequencing of SRPIN340-treated NIKS16 cells reveals reversal of some of the effects of HPV16 infection on keratinocytes.** (A) Heatmap of overall changes in gene expression caused by SRPIN340 treatment in NIKS16 cells differentiated in monolayer culture. Three replicates of DMSO-treated and 10 μM SRPIN340-treated NIKS16 cells were sequenced. (B) Pie chart showing the relative percentages of categories of SRPIN340 up-regulated genes in NIKS16 cells differentiated in monolayer culture. The GO terms for the up-regulated pathways are shown on the right-hand side. (C) Pie chart showing the relative percentages of categories of SRPIN340 down-regulated genes in NIKS16 cells differentiated in monolayer culture. The GO terms for the down-regulated pathways are shown on the right-hand side. (D) Graph showing genes involved in the epithelial structural barrier down-regulated by HPV16 infection [38] but up-regulated by SRPIN340 treatment in NIKS16 cells differentiated in monolayer culture. (E) Graph showing genes involved in innate

immunity oppositely regulated by HPV16 infection [38] and SRPIN340 treatment of NIKS16 cells differentiated in monolayer culture. (F) Over representation analysis showing the various cellular processes altered by SRPIN340-induced changes in alternative splicing in the above experiment.

**Table 1. Number of sequencing reads and percent alignment to the HPV16 and human genomes for each sample subjected to RNA sequencing.**

|  | Sample | No. of reads | Alignment to HPV16 genome (%) | Alignment to Human Genome (%) |
|---|---|---|---|---|
| No drug | 1 | 46578063 | 0.04 | 82.10 |
| No drug | 2 | 29538705 | 0.04 | 85.16 |
| No drug | 3 | 37758387 | 0.04 | 80.54 |
| SRPIN340 | 4 | 28468475 | 0.05 | 84.32 |
| SRPIN340 | 5 | 31260092 | 0.05 | 86.22 |
| SRPIN340 | 6 | 32025480 | 0.04 | 80.28 |

replicates of each condition (Fig 7). Read coverage of the human and viral genomes was not altered by drug treatment (Table 1). However, drug treatment caused a significant change in cellular gene expression (Fig 7A). A total of 342 cellular transcripts were up-regulated and 241 transcripts were down-regulated (S1 Table) with a fold change of>log 2.0, Q-value <0.05. Table 2 shows the top 20 up- and down-regulated genes. Ingenuity pathway analysis showed genes up-regulated by SRPIN340 treatment in HPV-infected cells to be involved in keratinocyte differentiation, metabolism, immune response and the epithelial barrier (Fig 7B). Down-regulated pathways included aspects of morphogenesis and cell motility. Cell adhesion and the immune system pathways were upregulated (Fig 7C) [25].

We compared the gene expression profiles of SRPIN340-treated NIKS16 cells to HPV16-induced gene expression changes identified in our dataset of gene expression changes due to HPV16-infection in NIKS cells [38]. For this analysis we used genes exhibiting a fold change>log 1.5 to capture the maximum number of genes in common. Analysis of these changes revealed 91 genes whose expression was altered in both conditions. Previously we showed that HPV16 infection leads to loss of both structural and immune aspects of the epithelial barrier [38]. Genes encoding proteins involved in keratinocyte terminal differentiation such as filaggrin and keratin 10, epithelial structural barrier such as loricrin, transglutaminase and repetin [39], and the immune barrier such as keratin 16 [40] and SPRR proteins [41] (Fig 7D), were up-regulated by drug treatment. Moreover, SRPIN340 affected the expression of six genes encoding key immune regulators (Fig 7E). SRPIN340 may have the potential to restore some aspects of normal keratinocyte function altered by HPV16 infection.

## SRPIN340 induces alternative splicing changes related to key processes in the HPV life cycle

RNA-Seq data from triplicate 10 μM SRPIN340-treated and DMSO-treated NIKS16 cells were aligned to the human genome. Alternative splicing analysis used the bioinformatic tool SplAdder. 'Percentage spliced in' (PSI) values are calculated based on the frequency of a particular exon being spliced into a specific splice isoform. PSI values calculated by SplAdder were used as a measurement of alternative splicing events comparing between SRPIN340-treated and DMSO-treated cells. Differentially spliced genes were then ranked by statistical significance. SRPIN340 treatment resulted in changes in alternative splicing of 935 genes (S2 Table).

**Table 2. Fold change of the top 20 up- and 20 down-regulated protein-coding genes by SRPIN340 treatment compared to DMSO treatment in NIKS16 cells. Q-value adjusted for multiple hypotheses by DESeq2 analysis.**

| Gene | Negative fold change | Positive fold change | Q-value |
|------|---------------------|---------------------|---------|
| TMPRSS11D | | 6.797647766 | 8.66E-05 |
| SERPINB11 | | 6.032431007 | 0.000679 |
| CYP1A1 | | 5.968312781 | 1.21E-25 |
| HRNR | | 5.415118392 | 0.004903 |
| ENDOU | | 4.90915265 | 1.9E-06 |
| LOR | | 4.607461151 | 0.001384 |
| FAIM2 | | 4.555164059 | 0.010715 |
| LIPN | | 4.437793821 | 0.017848 |
| SH2D1B | | 4.393057631 | 2.56E-05 |
| CRNN | | 4.163890496 | 6.37E-29 |
| CAPN14 | | 4.084787111 | 1.06E-05 |
| AQP5 | | 4.069593091 | 6.13E-19 |
| KLK12 | | 4.00172104 | 4.12E-13 |
| TMPRSS11A | | 3.98695018 | 3.71E-15 |
| FLG2 | | 3.887629517 | 9.25E-17 |
| CEACAM5 | | 3.819780979 | 0.000152 |
| S100A7 | | 6.797647766 | 8.66E-05 |
| ARG1 | | 6.032431007 | 0.000679 |
| SPINK7 | | 3.463702469 | 1.75E-31 |
| SPAG17 | | 3.4159458 | 8.97E-05 |
| ANKRD1 | -4.307959272 | | 0.000146 |
| MYL9 | -4.205082349 | | 6.30E-05 |
| MYL7 | -3.538350233 | | 2.91E-06 |
| GLP2R | -3.287298644 | | 0.021162 |
| MGP | -2.883346003 | | 5.85E-19 |
| KDR | -2.856988802 | | 3.51E-24 |
| ALDH1L2 | -2.742967901 | | 0.000597 |
| TGM2 | -2.707864354 | | 4.79E-17 |
| DRD2 | -2.678088804 | | 0.007043 |
| ALOX5AP | -2.67413198 | | 6.89E-06 |
| TAGLN | -2.544919939 | | 5.02E-34 |
| COL8A1 | -2.329975942 | | 1.59E-12 |
| SELENOP | -2.313804905 | | 0.035668 |
| TOX2 | -2.279567504 | | 0.035591 |
| WFDC2 | -2.742967901 | | 0.000102 |
| SLC7A7 | -2.087552302 | | 6.50E-05 |
| ANPEP | -1.964638827 | | 0.003945 |
| C3 | -1.951676084 | | 1.11E-30 |
| PTPRB | -1.933491109 | | 0.00429349 |
| SLCO2A1 | -1.915283426 | | 1.33E-03 |

RNAs identified as alternatively spliced were subject to pathway analysis to infer the biological significance of the changes induced by SRPIN340 treatment in NIKS16 cells. Over-representation analysis was performed on the 935 significantly alternatively spliced genes (p-value <0.05) by Webgestalt [42]. The biological processes affected by the changes in

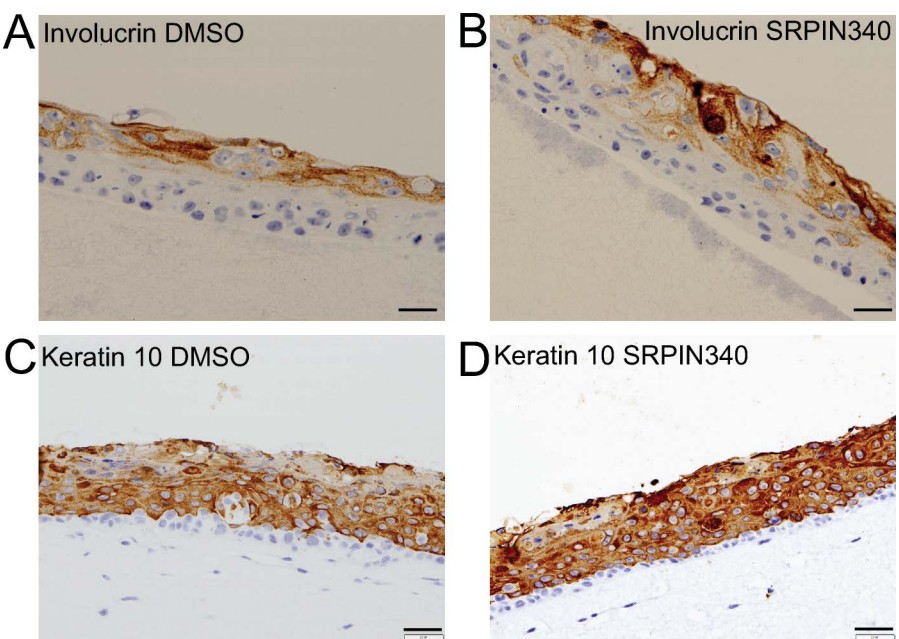

**Fig 8. SRPIN340 treatment enhances expression of epithelial differentiation markers.** 3D differentiated NIKS16 raft tissue sections immunohistochemically stained and treated as follows: A. Topically treated with DMSO for 48 hours. Stained with an antibody against involucrin. B. Topically treated with SRPIN340 for 48 hours. Stained with an antibody against involucrin. C. Topically treated with DMSO for 48 hours. Stained with an antibody against keratin 10. D. Topically treated with SRPIN340 for 48 hours. Stained with an antibody against keratin 10. Size bars=20μm.

alternative splicing caused by SRPIN340 treatment included RNA splicing, as expected due to the major role of SRPK1 in regulating splicing. However, splicing of transcripts encoding DNA repair factors/DNA damage response was also significantly altered (Fig 7F).

To confirm the RNASeq data showing expression of keratinocyte differentiation markers was enhanced by SRPIN340 treatment of HPV16-infected cells, we detected involucrin and keratin 10 differentiation markers by immunohistochemistry in sections of 3D differentiated NIKS16 raft tissues (Fig 8). Involucrin was detected in the upper layers of NIKS16 raft tissues topically treated with DMSO or with 10 μM SRPIN340. However, more involucrin appeared to be expressed in the SRPIN340-treated tissues compared to the DMSO-treated control (Fig 8B). Keratin 10 was detected in the mid to upper layers of DMSO-treated NIKS16 tissues (Fig 8C). There was an increase in keratin 10 expression in SRPIN340-treated tissues and expression of keratin 10 appeared to commence lower down the epithelium (Fig 8D).

## SRPIN340 is not toxic to keratinocytes

If SRPIN340 can inhibit HPV16 replication it could have therapeutic potential. However, cellular toxicity due to effects on host cell alternative splicing must be considered. We examined the toxicity of SRPIN340 in undifferentiated, uninfected NIKS and undifferentiated HPV-infected NIKS16 keratinocytes. Drug concentrations of 10 μM and 50 μM were tested over a period of 72 hours using cells grown in monolayer culture. DMSO was used as the vehicle control. Cell proliferation was not affected by either drug concentration in either cell line (Fig 9A, B). A small decrease in proliferation of NIKS16 cells was observed at the highest drug concentration following treatment for 72 hours. This was not statistically significant (Fig 9B). Cell viability of NIKS cells was only decreased significantly at the highest drug concentration

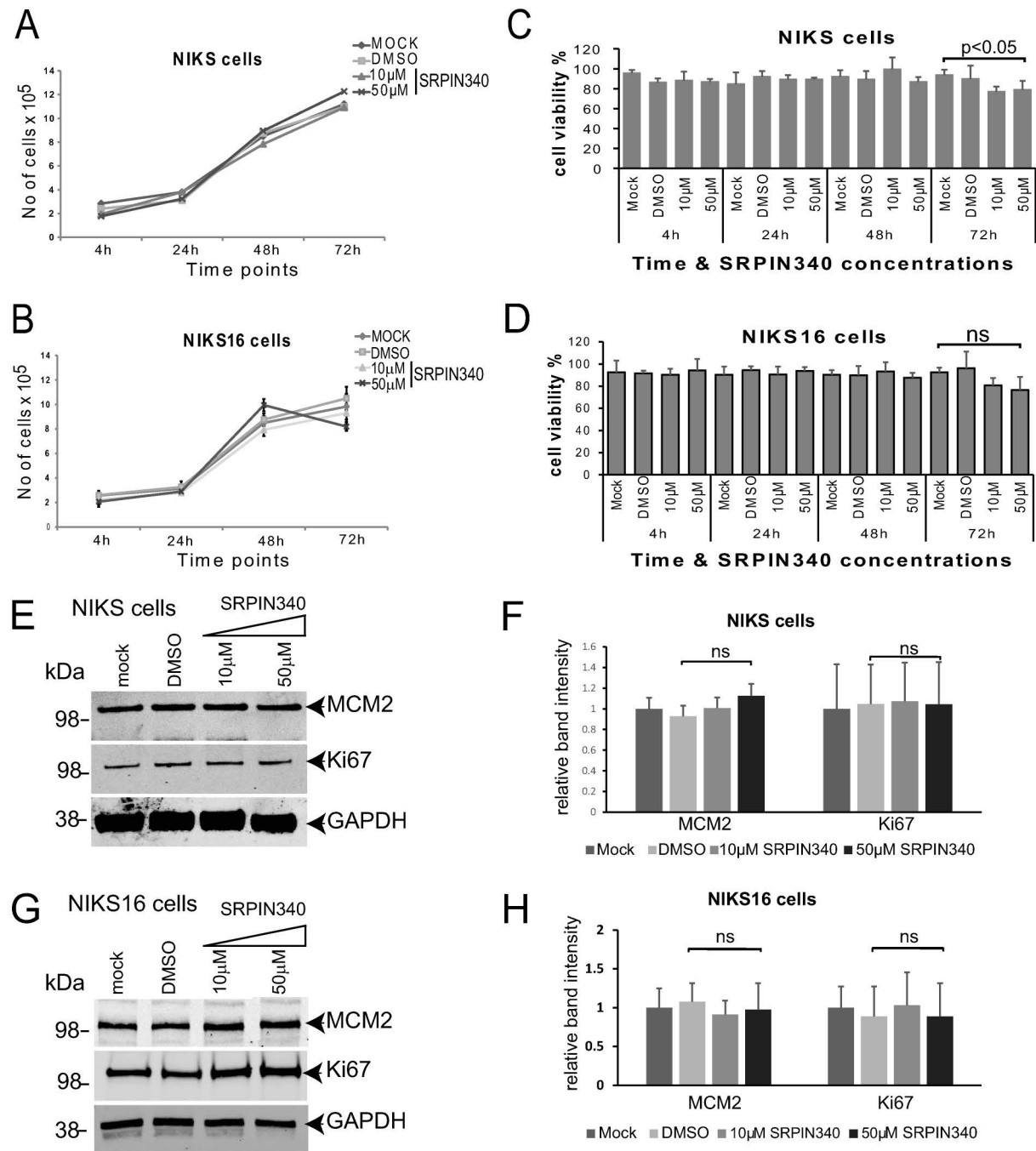

**Fig 9. SRPIN340 treatment does not alter cell proliferation or viability in monolayer cultured keratinocytes.** (A) Proliferation of undifferentiated HPV-negative NIKS cells in monolayer culture and mock-treated or treated with the drug vehicle DMSO or with 10 μM or 50 μM SRPIN340 over 72 hours. (B) Proliferation of undifferentiated NIKS16 cells in monolayer culture and mock-treated or treated with the drug vehicle DMSO or with 10 μM or 50 μM SRPIN340 over 72 hours. (C) Viability of HPV-negative monolayer cultured NIKS cells mock-treated or treated with the drug vehicle DMSO or with 10 μM or 50 μM SRPIN340 over 72 hours. (D) Viability of monolayer cultured NIKS16 cells mock-treated or treated with the drug vehicle DMSO or with 10 μM or 50 μM SRPIN340 over 72 hours. p<0.05, p-value of significance. (E) Western blot analysis of levels of MCM2 and Ki67 in HPV-negative undifferentiated monolayer cultured NIKS cells mock-treated or treated with the drug vehicle DMSO or with 10 μM or 50 μM SRPIN340 for 48 hours. GAPDH was used a loading control. (F) Graph of quantification of levels of MCM2 and Ki67 relative to GAPDH in western blots of protein extracts from monolayer cultured HPV-negative NIKS cells. The data shown are the mean and standard deviation from the mean from three separate western blot experiments. ns, no significant difference. (G) Western blot analysis of levels of MCM2 and Ki67 in undifferentiated monolayer cultured NIKS16 cells mock-treated or treated with the drug vehicle DMSO or with 10 μM or 50 μM SRPIN340 for 48 hours. GAPDH was used a loading control. (H) Graph of quantification of levels of MCM2 and Ki67 relative to GAPDH in western blots of

protein extracts from monolayer cultured NIKS16 cells. The data shown are the mean and standard deviation from the mean from three separate western blot experiments. ns, no significant difference.

following treatment for 72 hours (Fig 9C), while NIKS16 cell viability was unaffected (Fig 9D). Quantification of two cell proliferation markers, Ki67 and MCM2 showed no statistically significant change in cellular proliferation upon drug treatment of either NIKS (Fig 9E,F) or NIKS16 cells (Fig 9G,H). These data suggest that short-term SRPIN340 treatment is not toxic to HPV-negative or HPV-positive keratinocytes. However, we cannot exclude a significant change to cell proliferation or viability at later time points.

Finally, drug inhibition of replication could lead to viral genome integration. This experiment was carried out upon revision of the manuscript. The cells used for the other experiments in the manuscript (S1 Fig) were unavailable for this experiment. The monolayer-cultured and differentiated NIKS16 cells used in this experiment had a significant number of integrated genomes. S2 Fig shows droplet digital PCR analysis of relative numbers of total (S2A Fig) and episomal copies (S2A Fig) of the HPV16 genome in NIKS16 cells differentiated in monolayer culture and treated with either DMSO or treated with SRPIN340 for 48 hours. Using this accurate technique, we found no significant change in the relative numbers of episomal versus integrated genomes upon drug treatment in two separate experiments.

## Discussion

We have demonstrated that SRPK1 activity is essential for the expression of the HPV16 E2 replication/transcription factor at the mRNA and protein levels. Drug treatment did not affect the production of early spliced mRNAs or the late unspliced read-through mRNA containing open reading frames for L2 and L1. Therefore, SRPIN340 specifically inhibits the splicing which produces the E2 mRNA but does not generally affect viral transcription.

The question remains why SRPK1 had a specific effect on E2 mRNAs and did not disrupt all viral splicing. Unfortunately, it is not known what SRSFs control E2 mRNA splicing. Information on SRPK1 phosphorylation of SR proteins other than SRSF1 is lacking. Moreover, some SR proteins are predominantly phosphorylated by SRPK2 (e.g. SRSF2). Other SR proteins (e.g. SRSF1) are excellent substrates of SRPK1 but can also be phosphorylated by SRPK2 [43]. Therefore, SRPIN340 inhibition would differentially affect splicing through the relative importance of SRPK1 in phosphorylating different SR proteins. In HPV16 subgenomic plasmids transfected into HeLa cells, SRSF1 (an excellent substrate of SRPK1[43]) can enhance splicing from splice donor 880 to splice acceptor 3358 to form the E1^E4 containing mRNAs [44]. This mRNA was not altered by SRPIN340 here. In fact, previous data from NIKS16 cells containing endogenous episomal HPV16 genomes demonstrated that SRSF1 did not have a significant role in E1^E4 expression [22]. SRSF9 has also been shown to control E1^E4 splicing in the subgenomic/HeLa cell system but how SRPK1 regulates this protein is not known. HPV16 early gene splicing is controlled by SRSF2 (a poor substrate of SRPK1 [43]) and SRSF3 [45] (can be dynamically dephosphorylated [46]). Therefore, early splicing is less likely to be affected by SRPK1. Since we do not know which SR proteins regulate E2 viral mRNA expression we cannot be sure that SRPK1 affects E2 splicing.

E2 regulates alternative splicing of cellular genes [47]. This may be due to E2 activating transcription of SR proteins [22]. E2 can also interact with SR proteins to inhibit splicing of suboptimal introns [24]. HPV1 E2 is phosphorylated by SRPK1 [48] but although HPV16 E2 is subject to phosphorylation, it has not been shown to be a substrate of SRPK1. However, the close association of E2 with viral and cellular splicing suggests an important E2-splicing factor regulatory axis in the HPV-infected cell. The requirement for SRPK1 to produce E2 protein and E2 up-regulation of SRPK1 in differentiated epithelial cells [23] underscores this axis.

Surprisingly, although SRPIN340 treatment did not inhibit splicing of other major HPV16 mRNAs, it did reduce the cellular levels of HPV major late proteins E4 and L1. SRPK1 is present in the cytoplasm of differentiated HPV16-infected keratinocytes [23]. SRPK1 could indirectly control viral mRNA stability or translation in the cytoplasm by its known interactions in intracellular signalling pathways [49]. Alternatively it could phosphorylate cytoplasmic SR proteins such as SRSF1, to regulate viral protein translation [11].

We also showed that SRPIN340 alters the transcriptome, and can alter alternative splicing, in HPV16-infected keratinocytes. However, SRPIN340 treatment did not result in global changes to the transcriptome. There were significant changes in expression of 583 genes and alternative splicing of 935 genes was altered. In experiments where the drug was applied systemically in rats, the animals remained healthy [28]. This would not be expected if mRNA splicing was generally disrupted. Therefore, SRPK1 inhibition must have selected effects on splicing. This may be due to selective phosphorylation of SRSF substrates by SRPK1.

In our RNA-Seq analyses, SRPIN340 treatment led to up-regulation of expression of keratinocyte differentiation genes with 31% of drug-induced gene expression affecting epithelial differentiation. Keratinocyte differentiation factors included desmoglein 1, a cell-cell junction protein involved in epidermal homeostasis [50], SPINK5 a serine protease inhibitor, kallikrein-12 a serine protease, [51], loricrin, repetin and transglutaminase all required for the cornified envelope [39]. Expression of several keratins and keratin fibre-binding filaggrin was up-regulated [52]. Finally, several of the SPRR proteins involved in epithelial barrier protection and immune defence were also up-regulated [38]. Taken together, the up-regulation of these genes could enhance the epithelial barrier thus counteracting its disruption as a result of HPV16 infection. In addition, SRPIN340 treatment of NIKS16 cells changed alternative splicing of RNAs encoding factors involved in cellular processes essential for the HPV life cycle [32]. The effects of SRPIN340 on host cell gene expression may provide a host cell environment nonconductive to HPV16 replication, life cycle and transmission.

SRPIN340 has been proposed as a therapy against cancers such as metastatic melanoma [53]17. SRPIN340 alters VEGF splicing and inhibits angiogenesis [25]. VEGF mis-regulation is a key player in HPV-associate cancer progression. Moreover, VEGF-A inhibitors are a current second-line treatment for cervical cancer. Therefore, although we have studied SRPIN340 activity in the HPV16 life cycle, it may have potential in therapeutic treatment of HPV-associated cancers.

There are several limitations to this study. We carried out the analyses in differentiated HPV16-infected, monolayer cultured cells and in differentiated raft cultured tissues. Therefore, we cannot make any conclusions about the effect of SRPIN340 on early events in the viral life cycle or on persistently infected cells. Additionally, we did not include uninfected keratinocytes treated with SRPIN340 in our transcriptome analyses. Therefore, we cannot conclusively determine if the observed gene expression changes upon SRPIN340 treatment are specific to HPV16-infected cells or represent general effects on keratinocytes. Moreover, it would be important in future to determine if SRPIN340 could alter expression of E2 from other high-risk HPVs or low-risk genotypes causing warts. Our experiments were carried out only over two to three days. The long-term effects of SRPK1 inhibition on epithelial cell function warrant exploration. Sustained SRPK1 inhibition could impact processes such as cellular differentiation and tissue repair. Understanding these effects will be essential for developing safe and effective therapies targeting SRPK.

## Conclusions

We have identified host cell gene expression changes following SRPIN340 treatment in HPV16 infected cells. The anti-viral effect of SRPIN340 may be a combination of down-regulation of production of the viral E2 replication/transcription factor, together with changes

to splicing of host cell transcripts. These events collectively may alter the host cell environment to be inhibitory to late events in the HPV16 life cycle. The inhibition of HPV16 replication by SRPIN340 reveals SRPK1 to be a host factor crucial to the HPV life cycle.

# Materials and methods

## Cell culture

NIKS cells are normal spontaneously immortalised keratinocytes. NIKS16 cells are those cells transfected with HPV16 genomes [33,34]. NIKS and NIKS16 cells were grown in monolayer culture on a feeder layer of mitomycin-C-treated J2 3T3 cells as previously described [54]. They were used at a maximum passage number of 15 to avoid genome integration. NIKS and NIKS16 cells were induced to differentiate by growth for ten days at a high density in the presence of 1.2mM $Ca^{2+}$ [22]. Cells were maintained in 5% $CO_2$ at 37°C. NIKS and NIKS16 cells were treated with either DMSO or SRPIN340 (Sigma, UK: SML1088) dissolved in DMSO at the given concentrations on day 8 post-seeding into culture plates. For partially differentiated cells in E2 experiments, cells were treated on day 6 post seeding. Cells were incubated in the presence of the drug for 48 or 72 hours.

Raft cultures were established as previously described [55] by seeding NIKS16 cells onto the top of collagen matrices in 24 well plates. These were incubated at 37°C in 5% $CO_2$ for 2 days in E-medium until a monolayer was formed. Rafts were lifted to the air-liquid interface on a sterile stainless-steel wire grid and cultured for 14 days. The E-medium was changed every other day. DMSO or SRPIN340 (Sigma Aldrich, UK catalogue # 504293) dissolved in DMSO was added to the culture medium or placed on top of the raft tissues in a 50 µl droplet at the described concentrations on day 12 following lifting to the air-liquid interface and incubated for 48 hours.

## Cell growth kinetics, viability and tissue morphology

Formalin-fixed tissue cultures were paraffin-embedded and sectioned (2.5 µm sections). Sections were haematoxylin and eosin-stained and morphology was examined under an Olympus Bx57 light microscope. For cell viability and estimation of cell number, cells were grown in monolayer culture to give an undifferentiated population. Cells were harvested by trypsinization, diluted in 0.1% trypan blue at a ratio of 1:1, and counted using a haemocytometer. The number of unstained (viable) and total number of cells were determined.

## Tissue permeability of Alexa fluor 488

Cell to cell gap junction connectivity and raft permeability was assessed by the ability of cells to transfer the Alexa Fluor 488 dye (Invitrogen, UK, catalogue# A10436,). Alexa Fluor 488 dye at 10 µM was added to the top of 3D cultured tissues in a 50 µl droplet on day 12 of tissue growth and allowed to spread through the tissue structure for 2 days. At day 14, tissues were fixed, sectioned and counterstained with DAPI. Dye spread between neighbouring cells from the top to the bottom of the raft structure was visualized using a Zeiss LSM 710 confocal microscope. Zen black software (Zeiss) was used for capturing images.

## Southern blotting

DNA was isolated using the QiaAmp DNA mini kit (Qiagen, Germany) according to the manufacturer's instructions then subjected to restriction enzyme digestion overnight at 37°C with 25 units of enzyme per 20 µg DNA. To prepare the DNA probe, plasmid pEF-HPV16 was linearised by digestion with Bam HI at 10 units per µg DNA for one hour at 37°C. The

digestion products were fractionated on an agarose gel and the 8 kb viral genome band was excised and purified using the QIAquick gel extraction kit (Qiagen, Germany) according to the manufacturer's instructions. DNA was denatured by boiling for 5 min followed by chilling on ice. The denatured DNA was labelled with Biotin-16-dUTP (Merck, UK, catalogue #11093070910,) using the random primer labelling system according to the manufacturer's protocol, (Merck, UK, catalogue#11004760001). Restriction enzyme-digested NIKS16 DNAs and, as a size control, pEF-HPV16 digested with Bam HI, were fractionated on a 0.8% agarose gel in 1 x Tris borate electrophoresis buffer overnight at 20 volts. Gel treatment, DNA transfer onto nitrocellulose membrane and hybridisation with the pEF-HPV16 probe was carried out exactly as described in the Amersham Hybond N+ protocol (Cytiva, UK).

## Western blotting

Cells in monolayer were scraped into NP-40 lysis buffer (50 mM Tris-HCl pH 8.0, 150 mM NaCl, 0.5% (v/v) NP-40 (IGPAL CA-630)), containing PhosphoSTOP (Merck, UK catalogue # 04693116001) and complete miniprotease inhibitor cocktail (Merck, UK, catalogue # 200-664-3). Protein extraction from raft culture was carried out by adding a small amount of liquid nitrogen to the tissue then grinding to a powder in a mortar and pestle. NP-40 lysis buffer containing PhosphoSTOP (Roche) and complete miniprotease inhibitor cocktail as above was added to the ground tissue, mixed and collected in 0.5ml Eppendorf tubes. The insoluble matrix was removed by centrifugation for 20 min at 13,000 g at 4°C. All protein concentrations were determined using Bradford's assay and a NanodropTM spectrophotometer (Thermo Fisher Scientific, UK).

Protein samples were resolved by 4-12% Bis-Tris SDS-PAGE (Invitrogen, UK catalogue #NP0322BOX) using 1X NuPAGE MES gel running buffer (Invitrogen, UK, catalogue # NP0002). Separated proteins in the gel were electro-transferred onto nitrocellulose membrane at room temperature using the iBlot™ Dry Blotting System (Invitrogen, UK). Membranes were blocked in 5% foetal bovine serum (FBS) in phosphate buffered saline (PBS) for 60 min at room temperature. Membranes were incubated with primary antibody diluted in PBS-T (PBS with 0.1% Tween) containing 5% FBS overnight at 4°C before being washed three times with PBS-T. Primary antibodies were SRSF1 (1:1000, Mab96, Thermo Fisher Scientific, catalogue # 32-4500), SRSF2 (1:1000, Abcam, catalogue #Ab204916), SRSF3 (1:500, Life Technologies, UK, catalogue #334200), SRPK1 (1:500, clone G211-637 BD Transduction Laboratories, catalogue #611072), α-tubulin (1:5000, clone A11126, Thermo Fisher Scientific catalogue # 236-10501), involucrin (1:1000 clone SY5, Merck, UK, catalogue #19018), GAPDH (1:1000, Meridian Life Sciences, UK, catalogue #H86504M, clone 6C5), HPV16 E2 (1:500 Santa Cruz, Germany, catalogue #sc-53327, TVG271), HPV16 E6 (1:500, Santa Cruz, Germany, catalogue # sc-365089), HPV16 E7 (1:500 Santa Cruz, Germany, catalogue #sc-51951), HPV16 L1 (1:500 Dako, Denmark, clone K1H8), p53 (1:700, BD Pharmingen, UK, catalogue #554294), Rb (1:500, Cell Signaling Technologies, Germany, catalogue #9309, clone 4H1), MCM2 (1:500, Abcam, UK, catalogue #Ab133325), Ki67 (1:1000, Abcam, UK, catalogue #Ab197234), keratin 10 (1:500, Abcam, UK, catalogue #Ab9025). Monoclonal antibody104 (Mab104: detects phosphorylated SR proteins) was prepared from hybridoma supernatants (ATCC CRL-2067) and used neat. Secondary antibodies were diluted in PBS-T containing 5% FBS and incubated with the membrane for 1h followed by three washes in PBS-T, two washes in PBS, and one rinse in dH$_2$O. Secondary antibodies were goat anti-rabbit Dylight 800 conjugate (1:2000, Thermo-Fisher Scientific, UK, catalogue #SA5-35571), goat anti-mouse Dylight 800 conjugate (1:2000, Thermo-Fisher Scientific, UK, catalogue #SA5-35521) and IRDye anti-mouse 800CW (1:2000, IRDye Licor Biosciences Ltd, UK, catalogue #926-32210). Membranes were imaged

on an Odyssey Infrared Imager (LiCOR). The intensity of protein bands was quantified using Odyssey Image Studio software. Protein levels were determined and normalised to the level of the endogenous control.

## Immunofluorescence staining and confocal microscopy

Monolayer cultured cells were grown to 70% confluence on coverslips until differentiated. Coverslips were washed three times with PBS, Cells were fixed in PBS containing 5% formaldehyde at room temperature for 10 min. Permeabilization was carried out with PBS containing 0.5% NP40 at room temperature. Coverslips were then washed three times with PBS. Blocking was done in 10% filtered donkey serum in PBS for 30 min. This was followed by incubation with primary antibodies diluted in the blocking solution for 60 min at room temperature. Coverslips were washed three times with PBS followed by another wash in distilled $H_2O$. Primary antibodies were SRSF1 (1:250, Mab96, Thermo Fisher Scientific, catalogue # 324500), SRSF2 (1:500, Abcam, UK, catalogue #Ab204916), SRSF3 (1:300, Life Technologies, UK, catalogue #334200), HPV16 L1 (1:300 Dako, Denmark, clone K1H8), HPV16 E2 sheep antibody was a kind gift from Prof. Jo Parish, University of Birmingham, UK. It was used at a dilution of 1:300. HPV E4 antibody clone B11 was a kind gift from Prof John Doorbar, University of Cambridge. It was used at a dilution of 1:300. DAPI and secondary antibodies were diluted in blocking solution then added to the cells for 60 min, protected from light, prior to six washes in PBS followed by one wash in $dH_2O$. Secondary antibodies were Alexa fluor secondary anti-rabbit antibody 555 (1:1000, Invitrogen, UK, catalogue #A-31572), Alexa fluor secondary anti-mouse antibody 488 (1:1000, Invitrogen, UK, catalogue #A-21202), and Alexa fluor secondary anti-sheep 488 antibody (1:1000, Invitrogen, UK, catalogue #A-11-15). Coverslips were mounted on glass slides with a glycerol-based mounting medium (Citifluor, AF1) and sealed with nail enamel. Samples were examined using a Zeiss LSM 710 confocal microscope and Zen black software (Zeiss) was used for capturing images. Immunofluorescence staining of 3D organotypic raft culture sections was carried out on paraffin-embedded, fixed 2.5 μm sections. Antigen retrieval of the fixed section was performed by the histopathology department, Veterinary Diagnostic Services, University of Glasgow. Heat induced epitope retrieval (HIER) was carried out using the Menarini Access Retrieval Unit method.

## Immunohistochemistry

3D organotypic raft culture sections were prepared and antigen retrieval carried out exactly as for sections subject to immunofluorescence. For antibody staining, slides were loaded on to a Dako Autostainer, rinsed in Tris buffered saline containing 0.1% Tween (TBS-T), and blocked in Dako Real TM peroxidise blocking solution for 5 min followed by washing in TBS-T. Slides were incubated for 30 min in primary antibodies diluted in Dako universal diluent, 1/6000 for involucrin antibody, (Sigma Aldrich, UK, catalogue #19018) and 1/400 for keratin 10 antibody (Abcam, UK, catalogue # Ab9025). Following two washes in TBS-T, slides were incubated with secondary antibody for 30 min (Dako UK, catalogue #K4001) and washed twice in TBS-T. Slides were incubated for 10 min in diaminobenzidine (DAB) (Dako UK, catalogue #K5007) followed by three washes in water. Slides were counterstained with Gills Haematoxylin, dehydrated, cleared and then mounted in synthetic resin. Slides were examined under an Olympus Bx57 light microscope and images were captured using a x40 lens.

## RNA extraction and cDNA synthesis

Total mRNA extraction was performed using TRIZOL reagent (Invitrogen, UK) according to the manufacturer's instructions. For cells in monolayer culture, medium was removed, and

cells were washed twice with PBS. Cells were lysed by scraping into 500 μl TRIZOL reagent for each well of a 6 well plate. Purified RNA was dissolved in DEPC-treated $H_2O$ and stored at -20ºC. SuperScript™ III First-Strand synthesis system for RT-PCR (Invitrogen, UK) was used for cDNA synthesis according to manufacturer's instructions using $(dT)_{20}$ primers.

## End point RT-PCR

For a single round of amplification 500 ng of cDNA was used in a total volume reaction of 20 μl using 200nM primers, 200 μM dNTPs, 1.5 mM $MgCl_2$ and 2 units *Taq* polymerase (Invitrogen, UK). The amplification protocol was one cycle at 94º C for 2 min followed by 35 cycles of 94º C for 30 sec, annealing temperature of 55 º C (E4^L1: E4 Forward primer 5'-GTTGTTGCACAGAGACTCAGTGG-3' with L1 Reverse primer 5'-CGTGCAACATAT TCATCCGTGC-3') or annealing temperature of 57.8 º C (E6E7: E6 Forward primer 5'-GAGAACTGCAATGTTTCAGGACCC-3' with E7 Reverse primer 5'- GAACAGATGG GGCACACAATTCC-3') for 45 sec followed by extension of 72º C for 1 min. There was a final extension cycle of 72º C for 10 min. GAPDH cDNA was amplified as an internal control (Forward primer 5'-AGGAAATGAGCTTGACAAAG-3' and reverse primer 5'-ACCACAGTCCATGCCATCAC-3').

## Nested RT-PCR

Amplification of L2L1 readthrough mRNA and E2 spliced mRNA required two rounds of amplification. For L2L1, the first round of amplification used the same PCR reaction conditions described above with one cycle at 94º C for 2 min followed by 35 cycles of 94º C for 30 sec, annealing temperature of 57 º C for 45 sec followed by an extension of 72º C for 1 min. There was a final extension cycle of 72º C for 10 min (L2 forward primer 5'-ACATGCAGCC TCACCTACTT-3' with L1 reverse primer 1 5'-AACACCTACACAGGCCCAAA-3'). For the second round of amplification, 1μl of the first round PCR product was diluted in a total volume of 60 μl containing 200nM primers, 200 μM dNTPs, 1.5 mM $MgCl_2$ and 2 units *Taq* polymerase. The PCR conditions were one cycle at 94ºC for 2 min followed by 25 cycles of 94ºC for 30 sec, annealing at 57ºC for 45 sec and 72 ºC extension for 1 min. This was followed with a final extension of 72 ºC extension for 10 min (L2 forward primer 5'-ACATGCAGCC TCACCTACTT-3' with L1 reverse primer 2 5'-TGTCCAACTGCAAGTAGTCTGGATGT TCCT-3'). For E2 cDNA the conditions for the first round of amplification were the same as for L2L1 cDNA except using E1 Forward primer 5'-ATCTACCATGGCTGATCCTGC-3' and E2 Reverse primer 5'-GCAGTGAGGATTGGAGCACTGTC-3'. The conditions for the second round of amplification were the same as for L2L1 cDNA except that the annealing temperature was 63.2 ºC for 35 cycles of amplification using E1^E2 cross splice epitope Forward primer 5'-TGGCTGATCCTGCAGATTC-3' and E2 Reverse primer 5'-CTTCGGTGCCCC AAGGCGACGGCTTTGGTAT-3'. 10 μl of the final PCR products were fractionated on a 6% acrylamide gel. Electrophoresis was performed at 100 V and the gel was stained with 0.5 μg/ml EtBr for 15 min. PCR products were visualized under UV.

## Exonuclease V assay

Exonuclease V treatment of DNA was carried out as follows: 2 μl of NEBuffer 4 (10x), 2 μl of 10 mM ATP, 1 μl of Exonuclease V (New England Biolabs, UK, #RecBCD), 2 μl DNA (~200 ng) and 13 μl of $dH_20$ were mixed in a reaction tube. Reactions were incubated at 37°C for 1 hr followed by 95°C for 10 min. Samples were then stored at −20 °C until required. Equal amounts of treated and untreated NIKS16 DNA (40ng) were analysed by droplet digital polymerase chain reaction (ddPCR) to determine copy number. ddPCR was performed exactly

as described [56]. The RPP30 endogenous control probe primer set was used as an internal control. HPV16 E6-specific primers and probes were used at a final concentration of 300 nM. HPV16 E6 Forward Primer 5′-CAATGTTTCAGGACCCACAG-3′, E6 Reverse Primer 5′-CTGTTGCTTG CAGTACACACATTC-3′, E6 Probe 5′−CCACAGTTATGCACA GAGCTGC-3′) [56]. Reactions were mixed with Droplet Generation Oil on DG8 cartridges in the QX200 droplet generator (Bio-Rad, UK) to generate droplets. Thermal cycling conditions were: 95°C for 10 min followed by 40 × 30s at 94°C and 60°C for 1 min prior to final extension at 98°C for 10 min.

## Statistical analysis

Microsoft Excel software was used for plotting the data, calculating mean, standard deviations (SD), and p-values. Statistical significance was determined by students t-test with $p \leq 0.05$ considered significant. All experiments were performed at least three times.

## Next generation sequencing

RNA was prepared using the Qiagen RNeasy kit (Qiagen, Germany) from monolayer cultured NIKS16 cells grown for eight days to high density in the presence of 1.2. mM $Ca^{2+}$ followed by treatment with either DMSO or 10 μM SRPIN340 dissolved in DMSO for 48 hours. RNA quantification and quality was assessed by Qubit and Agilent TapeStation, respectively. Ribosomal RNA was depleted using the Illumina RiboZero Gold H/M/R Kit (Illumina, UK). Libraries were prepared using the TruSeq Stranded mRNA Kit (Illumina, UK). cDNA was made using a Superscript II reverse transcriptase reaction (Invitrogen, UK). cDNA was converted to dsDNA. It was cleaned up using Beckman Coulter's RNAClean XP (1.8 ratio) and then A-tailed. Illumina TruSeq LT adapters were ligated to the samples then cleaned up with a 1:1 bead ratio of AMPure XP followed by size selection using SPRIselect (Beckman Coulter, UK). The library was assessed for quality by Qubit and Agilent TapeStation. RNA paired-end sequencing was carried out using Illumina Genome Analyzer sequencing at the University of Glasgow.

## Alignment of fastq files and read summarization

The six raw RNA sequencing fastq files (3 x SRPIN340-treated, 3 x DMSO-treated) were analysed for quality using FastQC (version 0.11.2). Reads were trimmed of adaptor sequences and low-quality bases using Trimgalore. The trimmed reads were aligned to the human genome GRCh38 (*Ensembl*) using Hisat2 (version 2.0.5) [57] in the Linux environment (version 18.04.3). FeatureCounts (Version 1.5.1) [58] was used to quantify the reads by read summarization.

## Differential gene expression analysis

Differential gene expression analysis of SRPIN340-treated and DMSO-treated NIKS16 cells was determined using the Bioconductor package (Version 3.12) and DESeq2 (Version 1.30.1) [59], in the R environment (version 4.0.5). Differentially expressed genes were filtered by Q-value<0.05 with a log fold change cut off of 2 except for comparison of SRPIN340-related versus HPV16 infection-related changes to NIKS16 gene expression where log fold change of 1.5 was used. Genes which were differentially expressed were variance stabilisation transformed for normalisation and visualisation on a heatmap. Pheatmap (version 1.0.12) was used for visualisation.

## Pathway analysis

BiomaRt (version as 2.46.3) was used to convert *ensembl* gene ID into gene symbols before pathway analysis was performed. Over representation analysis was performed using

ClusterProfiler (version 3.18.1) [60] in the R environment. Enrichment of gene ontology (GO) terms was determined for the subontology category, biological processes for 354 up and 241 down regulated genes (log fold change >2). The statistical test used includes a hypergeometric statistical test with Bonferroni correction. The thresholds used were Q-value < 0.05. Functional clustering of GO terms was performed using enrichplot (version 1.12.0).

## Comparison of differentially expressed genes upon HPV16 infection

Differentially expressed genes from HPV16-infected NIKS16 cells and uninfected NIKS cells [38] were compared manually to differentially expressed genes upon SRPIN340 treatment. Genes involved in epithelial barrier function and innate immunity were identified from both data sets.

## Over representation analysis of alternative splicing events

Bam files from the RNA paired-end sequencing of 10 μM SRPIN340-treated or DMSO-treated NIKS16 cells were sorted by co-ordinate, indexed and subject to SplAdder analysis [61] to measure and quantify alternative splicing events. Percentage spliced in (PSI) values were quantified for each splicing event. An independent two-tailed students t-test was performed for DMSO-treated and SRPIN340-treated values to determine the most significant differentially spliced genes upon SRPIN340 treatment. Pathway analysis was performed using Webgestalt ((http://www.webgestalt.org/) [42]. Over representation analysis was carried out testing for biological processes using Benjamini-Hochberg multiple testing adjustment.

## Supporting information

**S1 Fig. NIKS16 cells contain episomal genomes and undergo late events in the viral life cycle.** (A) Southern blot analysis of HPV16 genomes in NIKS16 cells. DNA was isolated from monolayer cultured and differentiated NIKS16 cells and digested with either Bam HI, (cuts the genome once, lane 2), or Hind III, (does not cut the genome, lane 3). The HPV16 genome was cut from plasmid pEF-HPV16 with Bam HI as a size control for the episomal genome (lane 1). (B) Western blot analysis of HPV16 E2 and L1 expression in protein lysates from HPV-negative NIKS and NIKS16 cells grown in monolayer culture to give an undifferentiated cell population (U, lanes 1 & 2) or differentiated by growing to high density in the presence of 1.2 mM Ca$^{2+}$ (D, lanes 3 & 4). Involucrin and keratin 10 were used as controls for keratinocyte differentiation. GADPH is used as a loading control.
(TIF)

**S2 Fig. SRPIN340 treatment does not enhance viral genome integration.** The cells used for the other experiments in the manuscript (S1 Fig) were unavailable for this experiment. The monolayer-cultured and differentiated NIKS16 cells used in this experiment had a significant number of integrated genomes. The experiment was carried out using droplet digital PCR detection of E6 gene copies. Values are shown as genome copies/μl. A. Graph showing the average number of total HPV16 genomes comparing DMSO-treated to SRPIN340-treated cells. B. Graph showing the average number of episomal HPV16 genomes comparing DMSO-treated to SRPIN340-treated cells. The data shown represent two independent experiments.
(TIF)

**S1 Table. List of genes differentially expressed upon SRPIN340 treatment.** Differentially expressed genes were determined by DESeq2. Q-value <0.05.
(DOCX)

**S2 Table. List of genes where alternative splicing is significantly altered by SRPIN340 treatment.** Genes whose alternative splicing is altered by SRPIN340 treatment using PSI values determined by SplAdder analysis.
(DOCX)

## Acknowledgments

We would like to thank Prof. John Doorbar, University of Cambridge, UK, for providing NIKS and NIKS16 cells lines and the anti-E4 B11 antibody. The sheep anti-E2 antibody was a kind gift from Prof. Joanna Parish, University of Birmingham, UK. Our thanks go to Dr Jean Quinn, University of Glasgow, for 3D raft culture training.

## Author contributions

**Conceptualization:** Sheila V. Graham.

**Formal analysis:** Arwa A. A. Faizo, Clare Bellward, Hegel R. Hernandez-Lopez, Quan Gu.

**Investigation:** Arwa A. A. Faizo, Clare Bellward, Hegel R. Hernandez-Lopez, Quan Gu.

**Methodology:** Andrew Stevenson.

**Project administration:** Andrew Stevenson.

**Supervision:** Andrew Stevenson, Quan Gu, Sheila V. Graham.

**Validation:** Arwa A. A. Faizo, Clare Bellward, Hegel R. Hernandez-Lopez.

**Visualization:** Arwa A. A. Faizo, Hegel R. Hernandez-Lopez.

**Writing – original draft:** Sheila V. Graham.

**Writing – review & editing:** Arwa A. A. Faizo, Hegel R. Hernandez-Lopez, Quan Gu, Sheila V. Graham.

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
