## [Decision Letter · Decision Letter 0]

4 Dec 2024

PPATHOGENS-D-24-02327

The splicing factor kinase, SR protein kinase 1 (SRPK1) is essential for late events in the human papillomavirus life cycle.

PLOS Pathogens

   Dear Dr. Graham,

Thank you for submitting your manuscript to PLOS Pathogens. After careful consideration, we feel that it has merit but does not fully meet PLOS Pathogens's publication criteria as it currently stands. Therefore, we invite you to submit a revised version of the manuscript that addresses the points raised during the review process.

Please submit your revised manuscript within 60 days Feb 02 2025 11:59PM. If you will need more time than this to complete your revisions, please reply to this message or contact the journal office at plospathogens@plos.org. Please include the following items when submitting your revised manuscript:

We look forward to receiving your revised manuscript.

Kind regards,

Denise A Galloway

Academic Editor

PLOS Pathogens

Robert Kalejta

Section Editor

PLOS Pathogens

 Sumita Bhaduri-McIntosh

Editor-in-Chief

PLOS Pathogens

orcid.org/0000-0003-2946-9497

 Michael Malim

Editor-in-Chief

PLOS Pathogens

orcid.org/0000-0002-7699-2064

**Additional Editor Comments:**

The reviewers felt that this manuscript provided interesting data to show that an inhibitor of SRPK1, SRPIN340, specifically targeted the HPV16 E2 gene expression and E2 plus E4 and L1 protein levels. To say that these changes are due to the drugs effects on SRPK1 it is necessary to show the defects in splicing of all the transcripts. What accounts for the difference between early and late gene splicing? Throughout the manuscript each figure needs to clearly indicate which cell culture conditions are being used. It is also not clear why HeLa cells are an appropriate control for NIKS16. Show controls for the E6 and E7 antibodies.

**Journal Requirements:**

At this stage, the following Authors/Authors require contributions: Arwa A. Faizo, Claire Bellward, Hegel Hernandez-Lopez, Andrew Stevenson, Quan Gu, and Sheila Graham. Please ensure that the full contributions of each author are acknowledged in the "Add/Edit/Remove Authors" section of our submission form.

2) We have noticed that you have uploaded Supporting Information files, but you have not included a list of legends. Please add a full list of legends for your Supporting Information files after the references list.

3) The following figures have poor resolution: S1, and S3, please provide higher resolution versions.

4) We note that your Data Availability Statement is currently as follows: "The raw data for the results described here is available upon request. RNA-Seq data sets are freely available from the European Nucleotide Archive accession number PRJEB80489.". Please confirm at this time whether or not your submission contains all raw data required to replicate the results of your study. Authors must share the “minimal data set” for their submission. PLOS defines the minimal data set to consist of the data required to replicate all study findings reported in the article, as well as related metadata and methods (https://journals.plos.org/plosone/s/data-availability#loc-minimal-data-set-definition).

- The points extracted from images for analysis..

**Reviewers' Comments:**

Reviewer's Responses to Questions

**Part I - Summary**

Reviewer #1: The authors examined the effects of an SRPK1 specific inhibitor SRPIN340 on HPV16 gene expression and life cycle in a cell line NIKS16 which harbors episomal HPV16 DNA. Experiments were conducted in submerged and differentiated (induced by high Ca++) cultures and in 3-D raft cultures. The authors also analyzed alternations in host gene expression and alternative RNA splicing based on using RNA seq data.

The authors presented some intriguing data, indicating that SRPK1 inhibited the expression of HPV16 E2 mRNA and E2 protein, as well as the production of the late E4 and L1 proteins. Interesting, the spliced E4 and L1 mRNAs were not affected. The transcription, RNA splicing and synthesis of the viral oncoproteins E6 and E7 were not affected. The authors proposed that this inhibitor could be a therapeutic target.

Reviewer #2: This report demonstrates that inhibition of SRPK1 disrupts viral splicing of E2 resulting in loss of E2 expression. This result supports the premise that SRPK1 inhibition is a target that could prevent HPV infections. The strengths of the paper are the potential significance of the results. The weaknesses are related to the NIKS system which is not an ideal model for HPV16 infections. However, overall there are strengths that could outweigh the weaknesses. A few points to consider from the results presented.

1. The author assignemts should be updated, a couple have no assignment. I assume that all of the work was done at UoG, therefore everyone should be assigned as UoG and current addresses can then be added.

2. Lines 175-177 state that Figs 1A-C were generated from proteins extracted from raft tissues. This is contradicted in the figure legend (lines 680-682). Please correct.

3. It isn't clear from this figure (1A-C) and the description what antibodies were used for what blot. pSRSF1 (and pSRSF3) while SRSF2 is listed. The legend states SRSF1 etc antibodies were used. The text is also confusing. Could the authors please clarify these figures so that they read better.

4. The E6 and E7 antibody blots lack any controls, there are no non-HPV samples included. There's no need to repeat everything, but demonstrating the specificity of the bands shown with a non-HPV sample is required. Also, the authors have used HPV18 HeLa extracts as a control for the antibodies. Do the authors know whether the antibodies used cross-react with 16 and 18? If they do that's fine, but stating there is more E6 and E7 in the HeLa cells should be removed as this may be due to antibody affinity.

5. Given that SRPK1 inhibition disrupts E2 expression during differentiation the authors should determine whether viral genome integration has occurred. Loss of E2 will disrupt viral genome replication and amplification during differentiation which could result in viral integration. There's no need for Southern blots to answer this question, TV exonuclease treatment of DNA harvested from cells will easily demonstrate a trend to integration. This would also help to explain the potential disruption in other viral proteins during differentiation. The dysplastic 3D tissue following SRPK1 inhibition would support the idea that there is viral genome integration.

Reviewer #3: The manuscript by Faizo and colleagues describes using a pharmacological approach to demonstrate the importance of the splicing regulator SRPK1 in the human papillomavirus lifecycle. More specifically they show that a small molecule inhibitor that inhibits SRPK1 activity reduces the levels of the E2 transcript and protein. E2 is a crucial virus encoded regulator of virus replication and gene expression and loss of E2 impacts on the late stages of the HPV lifecycle in a model of the virus lifecycle. RNA sequencing also shows that treatment with the small molecule inhibitor alters host cell gene expression, although it is not clear if this is only in the context of HPV infection or a broader effect. Crucially, many of the changes in gene expression correlate with targets of virus and likely counteract the de-differentiation mechanisms of the virus.

Overall this is a well written manuscript that is accessible to a wider audience. The experiments are generally well controlled and the figures are good quality.

**Part II – Major Issues: Key Experiments Required for Acceptance**

Reviewer #1: There are a number of concerns as listed below:

Abstract and Introduction ( Since both sections also summarized their results and conclusions, some comments are provided under Results and Discussion)

Lines 41,42, “Drug-mediated loss of E2 is specifically associated with inhibition of late gene expression.” However, the data showed neither transcription nor splicing of early and late viral genes were affected. Only the E2 mRNA and protein, as well as E4 and L1 proteins were drastically reduced. There were no experiments to shed light on the reason for the dramatic reduction in E4 and L1 proteins. Only speculations (see Discussion section).

Lines 48,49, 131,132. “The treatment was not associated with changes in cell proliferation or differentiation as determined in monolayer cultures.” Please qualify the conclusion by stating "short term treatment (2 or 3 days)".

Lines 44,45, ..”the loss of…. immune function was restored”. See comments below.

Results

Line 175-179. Fig. 1 A-C : The text indicated the experiments were conducted with raft cultures. However, the Legend stated NIKS16 cells were grown as monolayer cultures. Which culture was used?

Fig 1D-F: Clearly, monolayers cultures were used for IF, but there is no information on whether the cells were differentiated. (the same question Figs. 2 and 5, 6)

Line 175-179, the statement “SRSF1 and SRSF3 were found in the cytoplasm” is overstated; it appears that only portions of each protein were in the cytoplasm. Nuclear localization cannot be ruled out in the absence of subcellular fractionation.

Line 142-Fig. 3 Legend, HPV16 "late" promoter….. should be "early" promoter.

Line 266-270. “ One important known change to the epithelium caused by HPV 16 infection is an abrogation of keratinocyte differentiation and loss of both structural and immune aspects of the epithelial barrier.” This statement is not entirely correct and should be rephrased. The virus life cycle is dependent on epithelial differentiation (which the author also stated in the Introduction). Indeed, the involucrin expression is not affected as shown in the Figs. 3 and 4. As far as I am aware of, keratin 10 was not abrogated either.

Line 274,275: “SRPIN340 affected the expression of 5 genes encoding key immune regulators, including IRF1 (Fig .5E), which are known to be modulated by HPV16 infection.” Fig. 5E, shows that he HPV16 infected cells down-regulated 3 immune response genes, while upregulating 3 others, including IRF1. The figures also shows that SRPIN340 had the opposite effects on these 6 genes, including down-regulating IRF1. However, HPV oncoproteins are known to suppress immune response genes, including IRF1. (a similar concern for Statements in Discussion section as well)

Line 296-299, “Splicing of transcripts encoding DNA repair factors/DNA damage response was also significantly altered (Fig 5F). The DNA damage response is essential for HPV replication and alternative splicing-induced changes in DNA repair factors could create a cellular environment inhibitory to HPV replication.” This statement is an overreach without validation. It should be deleted. First of all, the loss of viral E2 protein would have incapacitated viral DNA replication. Secondly, the authors never examined viral DNA replication in inhibitor treated or control untreated raft cultures. Lastly, there are many DNA repair factors. The authors did not provide any information on the identity of these host factors with altered splicing. Therefore, it is not possible to know whether these factors are known to be required for viral DNA replication.

Fig .6. The author statement in the Discussion that they studied only differentiated NIKS16 in submerged cells. But this cannot be the case, as they can only examined cell proliferation and viability by using proliferating cells. It is recommended that authors explicitly specify the culture conditions for each experiment in the text and in the Fig. Legends.

In Fig. 6. The inhibitor appeared to reduce the cell number and viability at 72 hr. but the authors considered it insignificant. Only longer treatment can reveal whether the cell number and viability continued to trend down.

Discussion

This section was long and much was just speculation. It should be revised and shortened. Some examples and additional minor points are presented below.

Line 325,”SRPIN340 specifically inhibits alternative splicing of the E2 mRNA”. It is more accurate to state “…inhibits the splicing which produces the E2 mRNA.”

The authors speculated perhaps SRSF5 (and hnRNP G) might play roles in generating the E2 mRNA. Did the authors detect any changes in SRSF5 expression at the level of RNA or protein?

Line 342-343, “SRPK1 may be involved in promoting viral DNA replication, genome transcription, and mRNA splicing.” This statement is not completely accurate. The authors data show that the viral genome transcription was not affected, only the RNA splicing to produce the E2 mRNA was. SRPK1 is involved in promoting viral DNA replication, but only indirectly through its effect on RNA splicing to generate the E2 mRNA, hence protein.

Line 351-352 “ .. E2 can bind both E4 and L1 proteins [46, 47], could cause a reduction in levels of both proteins at a post-transcriptional level or perhaps by destabilising the proteins. This is purely illogical speculation. Since there is little or no E2 protein, how would it destabilize the late proteins? Did the author meant to say, the lack of E2 would destabilize the late viral proteins? This is very unlikely either. Chances are the early phase will not overlap the late phase for a simple reason. it is not possible to encapsidate viral DNA while it is undergoing replication. As to E4, it is the most abundant viral protein and is cytoplasmic, not nuclear as the E2 of low abundant. Their “transient interaction if any, or lack of it could not have affected the level of E4 protein.

Line 355- 358, “Inhibition of late events in the viral life cycle could imbalance the differentiation-dependent life cycle and potentially result in persistent infection and cancer progression. However, we found no evidence for any tumour-promoting activity of SRPIN340 in HPV16-positive keratinocytes, since cell proliferation and viability were unchanged in the presence of the drug, and repression of differentiation due to HPV infection was reversed. Importantly, we found that E6E7 expression and activity were not enhanced by drug treatment. “ . A 2 day treatment in raft cultures and 3-day treatment on submerged cultures cannot be used to predict its potential to promote cancer progression. This statement should be deleted.

Line 372-374, “This drug induced effect would be predicted to mechanically strengthen infected, differentiated keratinocytes and inhibit HPV virion egress from the surface of the epithelium.” This sentence is partially true. However, there will not be virus production for lack of L1 and for the inability to replication DNA in the absence E2.

Line 393-396,” Therefore, although we have studied SRPIN340 activity in the HPV16 life cycle, it may have potential in therapeutic treatment of HPV-associated cancers. Cancer-specific splicing can lead to the expression of oncogenic protein isoforms, as has been shown for HPV-positive HNSCC [51, 52], which SRPIN340 could ameliorate.” This manuscript did not address the issue whether the inhibitor can alter cancer-specific alternative splicing in host genes. The statement should be deleted.

Reviewer #2: 1. Demonstrate that the oncoprotein antibodies have specificity by running a non-HPV control side by side with HPV16 samples.

2. Determine whether SRPK1 inhibition promotes viral genome integration due to the disruption of E2 expression. This is important as integration is often associated with progression to cancer.

Reviewer #3: (No Response)

**Part III – Minor Issues: Editorial and Data Presentation Modifications**

Reviewer #1: The manuscript needs a thorough read to improve writing and clarity. For instance, there are many ….that…….that…; …that… which; …. whose….which….. Long and complex sentences should be split into sperate sentences.

Reviewer #2: See above.

Reviewer #3: In this reviewer's view the supplementary experiments should be moved (if possible) into the main manuscript as they are necessary for understanding the early section of the results chapter. Having them in the main body will help the reader.

Fig1D - in the DMSO merge it looks like there might be an issue with the image. It appears as though some of the cell are obscured by something. Has this image been manipulated?

Fig 2 – it is not clear to me why the authors chose to use HPV18 positive HeLa cells as the control for E6/E7 expression in these experiments rather than a HPV16 positive cancer cell line. Also, in the graphs it might be better to normalise to the NIKS16 mock rather than the cancer cells, which have much higher levels of oncogene expression. This might make it easier to interpret the data.

Fig 3 - These changes are not essential but the data in fig 3 could be strengthened through alternative approaches such as shRNA knockdown of SRPK1 or introduction of DN kinase mutations.

Fig 4 – The western blots and staining of the rafts show a very clear reduction in late viral protein expression. What is the mechanism for this given that no major differences in late transcript levels were seen in the previous figure?

Fig 5 – how do the gene expression changes relate to the previous studies showing E2 regulates gene expression changes?

Fig 5 - it would be important to confirm some of the gene expression changes seen in the RNA sequencing at protein level in your model systems. In particular it would strengthen the manuscript to show that levels of differentiation markers have been altered. Would you have anticipated being able to see changes in morphology of the treated rafts given the RNA-seq data?

What happens after longer term treatment of the drug? Do you see a greater impact on NIKS16 proliferation?

PLOS authors have the option to publish the peer review history of their article (what does this mean? ). If published, this will include your full peer review and any attached files.

**Do you want your identity to be public for this peer review?** For information about this choice, including consent withdrawal, please see our Privacy Policy .

Reviewer #1: No

Reviewer #2: No

Reviewer #3: No

**Figure resubmission:**
---

## [Editor Report · Decision Letter 1]

11 Mar 2025

Dear Prof Graham,

We are pleased to inform you that your manuscript 'The splicing factor kinase, SR protein kinase 1 (SRPK1) is essential for late events in the human papillomavirus life cycle.' has been provisionally accepted for publication in PLOS Pathogens.

Best regards,

Denise A Galloway

Academic Editor

PLOS Pathogens

Robert Kalejta

Section Editor

PLOS Pathogens

Sumita Bhaduri-McIntosh

Editor-in-Chief

PLOS Pathogens

orcid.org/0000-0003-2946-9497

Michael Malim

Editor-in-Chief

PLOS Pathogens

orcid.org/0000-0002-7699-2064

The authors have made reasonable corrections to address the reviewers concerns.
---

## [Editor Report · Acceptance letter]

Dear Prof Graham,

We are delighted to inform you that your manuscript, "The splicing factor kinase, SR protein kinase 1 (SRPK1) is essential for late events in the human papillomavirus life cycle.," has been formally accepted for publication in PLOS Pathogens.

Best regards,

Sumita Bhaduri-McIntosh

Editor-in-Chief

PLOS Pathogens

orcid.org/0000-0003-2946-9497

Michael Malim

Editor-in-Chief

PLOS Pathogens

orcid.org/0000-0002-7699-2064